# Multiphysics Modeling of a Synthetic Jet Actuator in Operation

**Matthew G. M. Butler** [1], **Alis Ekmekci** [2,†] and **Pierre E. Sullivan** [2,*,†]

1    Department of Mechanical and Industrial Engineering, University of Toronto, Toronto, ON M5S 3G8, Canada; mgm.butler@mail.utoronto.ca
2    Institute for Aerospace Studies, University of Toronto, Toronto, ON M3H 5T6, Canada; alis.ekmekci@utoronto.ca
*    Correspondence: sullivan@mie.utoronto.ca
†    These authors contributed equally to this work.

**Abstract:** Active flow control is a promising technology for reducing noise, emissions, and power consumption in various applications. To better understand the performance of synthetic jet actuators, a computational model that couples structural mechanics with electrostatics, pressure acoustics, and fluid dynamics is needed. The model presented here was validated against experimental data and then used to investigate the fluid behavior inside and outside the synthetic jet actuator cavity, the impacts of thermoviscous losses on capturing the acoustic response of the actuator, and the viability of different modeling methods of diaphragms in computational simulations. The results capture the feedback from the fluid onto the diaphragm and highlight the need for careful acoustic modeling.

**Keywords:** synthetic jets; flow control; multiphysics modeling

## 1. Introduction

Global efforts to reduce noise, emissions, and power consumption have led to rapid developments in active flow control—adding energy to a flow to change its aerodynamic characteristics. Passive flow control efforts, such as flaps on airplanes or slots on wind turbine blades [1], have been effective at improving efficiency, but active methods are required to achieve significant further improvements [2]. This will only become more important as technology such as aviation and wind power generation continue to grow in importance and popularity.

Synthetic jet actuators (SJAs) are simple active flow control devices with sizes generally on the order of millimeters. They typically comprise a cavity with one or more diaphragms mounted inside and an orifice or nozzle that leads to the surface where energy addition is desired. By applying a time-varying sinusoidal electric potential, the piezoelectric disk—clamped around its edges—flexes inwardly and outwardly, effectively shrinking and enlarging the cavity of the SJA. The movement of the diaphragm creates alternate ingestion and expulsion of fluid from the cavity (Figure 1).

These devices are advantageous because they can be implemented flush with a surface, introducing no parasitic drag. They are compact, inexpensive, and simple, and they require no external fluid supply. Their performance can also be manipulated by altering the voltage signal, meaning they can be tailored to different applications without requiring significant redesign. These advantages make synthetic jet actuators appealing devices for various fields.

A downside that SJAs present is their noise generation. Because of the vibration of the diaphragm and resonance of the cavity, some designs generate significant noise [3–5]. Excessive noise generation can be a critical design drawback, especially in applications such as transportation, where noise reduction is a focus. Arafa et al. [6] found that when designing SJA arrays, an excitation frequency away from the cavity acoustic modes should be used to ensure uniform mean jet velocity across the array. They achieved this target by

dividing the SJA cavity into isolated sections. Arafa et al. [7] revealed that it is possible to reduce the SJA noise by 8–10 dB by operating it at 60–80% of its Helmholtz frequency and still achieve penetration of comparably high jet momentum into the fluid outside the SJA. Furthermore, they showed that jet momentum injected through a rectangular slot may be high at the slot exit but does not travel as far into the fluid outside the SJA as the momentum injected through an array of circular orifices with comparable open area. Jabbal and Jeyalingam [5] aimed to reduce the noise output of a piezoelectric SJA by using a double-chamber actuator. The design reduced noise by 26% while reducing peak jet velocity by just 7%. Wang et al. [3] studied the fundamental sound generation mechanisms of SJAs at frequencies from 100 to 600 Hz. They used four separate experimental configurations to isolate the two monopole sound sources: the pulsating jet (fluid-borne noise) and the motion of the diaphragm (structure-borne noise). They found that at low frequencies, cancellation occurs between the two monopoles.

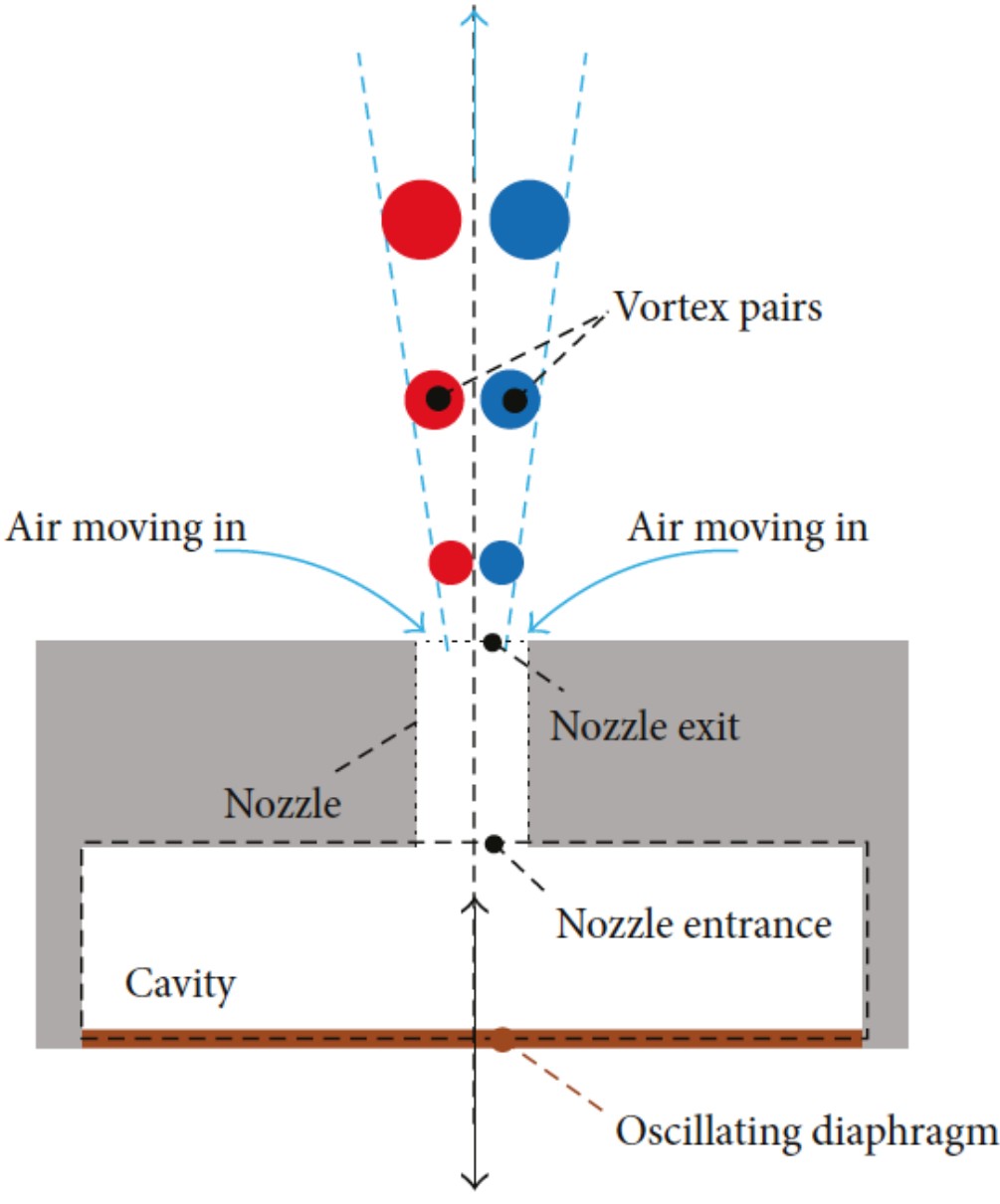

**Figure 1.** Cross-section schematic of an opposite-facing piezoelectric synthetic jet actuator [8].

The time- and space-averaged jet velocity ($\bar{U}_j$) at the exit of the SJA is

$$\bar{U}_j = \frac{1}{T/2}\frac{1}{A_j}\int_{A_j}\int_0^{T/2} u_j \mathrm{d}t \mathrm{d}A_n \tag{1}$$

where $T/2$ is half the period of oscillation (expulsion stroke), $A_j$ is the cross-sectional area of the orifice, and $u_j$ is the fluid velocity at a given point [9].

SJAs are often described by their jet Reynolds number and Stokes number (Equations (2) and (3), respectively), where $d$ is the orifice diameter, $\omega$ is the circular frequency ($\omega = 2\pi f$), and $\nu$ is the fluid's kinematic viscosity. The Stokes number describes the fluid response to the periodic motion.

$$Re_{\bar{U}} = \frac{\bar{U}d}{\nu} \tag{2}$$

$$S = \sqrt{\frac{\omega d^2}{\nu}} \tag{3}$$

Another important parameter that indicates a performance characteristic of an SJA is its Helmholtz frequency (the natural resonance frequency of its cavity with the neck). Equation (4) gives the Helmholtz frequency for an SJA with an axisymmetric orifice, where $A_n$ is the orifice cross-sectional area, $c$ is the speed of sound, $l_n$ is the nozzle length, and $V_c$ is the cavity volume [10].

$$f_H = \frac{1}{2\pi}\sqrt{\frac{3}{4}\frac{A_n c^2}{l_n V_c}} \tag{4}$$

Rampunggoon [11] proposed a jet formation criterion based on an order-of-magnitude analysis, which was validated computationally and experimentally [12,13]. Holman et al. [14] defined that jet formation occurs when $Re/S^2 > 0.16$ for the axisymmetric jet.

Early computational work by Kral et al. [15] simulated an SJA as an inlet velocity profile boundary condition (omitting the cavity and nozzle) using a 2-D Reynolds-averaged Navier–Stokes (RANS) approach in quiescent flow. They found that the laminar synthetic jet case did not agree with experimental results, but the turbulent case showed good agreement. Donovan et al. [16] performed 2-D unsteady RANS simulations of a synthetic jet on NACA airfoils in the presence of a cross flow. They also used a suction/blowing boundary condition with an oscillatory velocity profile modeled with an equation. Their simulations showed good agreement with experiments, which gave confidence that RANS could be used to estimate the effectiveness of synthetic jets with forcing frequencies near the vortex shedding frequency. They confirmed the finding that SJAs could be used to increase lift. Tang and Zhong [17] performed laminar and turbulent 2-D axisymmetric simulations using FLUENT and compared their results to experiments. The diaphragm was modeled with an equation as an oscillating velocity profile from the theory of plates and shells [18]. They found that FLUENT could match experimental results from their laminar simulations very well and that the RNG $k - \varepsilon$ and standard $k - \omega$ turbulence models performed best for the turbulent cases. Mane et al. [19] used the same diaphragm boundary condition to simulate two types of composite diaphragms, including THUNDER™(commercially available piezoelectric actuators and sensors). They found that the diaphragm displacement profile from Timoshenko [18] did not accurately represent the THUNDER actuator profile and, therefore, used a parabolic profile instead. Similarly, Jain et al. [20] reported poor agreement with experiments when simulating SJAs with a sinusoidally moving piston and an oscillating velocity profile with a top hat shape. Instead, they found that a moving wall with a parabolic shape yielded more favorable results. They also performed extensive simulations to determine the effect of altering cavity and orifice parameters. Rizzetta et al. [21], Ziade et al. [22], and Sharma [23] also performed computational studies where the diaphragm was modeled as a piston. Ho et al. [24] performed three-dimensional unsteady RANS computational studies of an SJA operating in a turbulent cross flow. They showed

how jet momentum affects the turbulent boundary layer and studied the turbulent structures that emanate downstream. Lumped element modeling (LEM) simplifies physical systems by approximating them with circuit elements concentrated at specific locations. Though several studies have shown success in modeling SJAs with LEM [23,25,26], complicated physics (such as the coupled interaction between the fluid, diaphragm, and acoustics) cannot be captured. Finally, Qayoum and Malik [27], Gungordu [28], and Gungordu et al. [29,30] performed simulations that incorporate multiphysics using COMSOL. These studies incorporated the exact solution of the diaphragm deflection in response to an electric potential, but only [28] included acoustic effects. The benefit of multiphysics modeling is creating a realistic computational model that factors in physics from multiple domains. This allows the diaphragm to be accurately modeled instead of approximated by an equation or velocity boundary condition.

The motivation for the work is to develop a computational model that couples all the relevant physics for piezoelectrically driven SJAs in one model. Modeling a group of physical parameters simultaneously in one model not only improves the understanding of SJA operation and optimization but also presents a more accurate representation of SJAs in general. This work demonstrates how diaphragm motion impacts the fluid flow as well as how flow impacts the motion of the diaphragm. Including pressure acoustics allows for extracting detailed acoustics information, such as sound pressure level and resonant peaks.

## 2. Materials and Methods

This work models the same SJA as in Feero et al. [31]. The chosen SJA geometry in this work is a cylindrical configuration with a single circular diaphragm positioned opposite the nozzle. The cavity and nozzle were both cylindrical, with diameters of 30.8 mm and 2 mm, respectively, and heights of 10 mm each.

The diaphragm used in the experiments was a THUNDER TH-5C piezoelectric transducer clamped around its edge by two plates. Like Feero et al.'s experiments [31], a time-varying voltage signal was applied to the transducer to induce diaphragm deflection. A cross-sectional depiction of the axisymmetric cylindrical SJA and its boundary conditions can be seen in Figure 2.

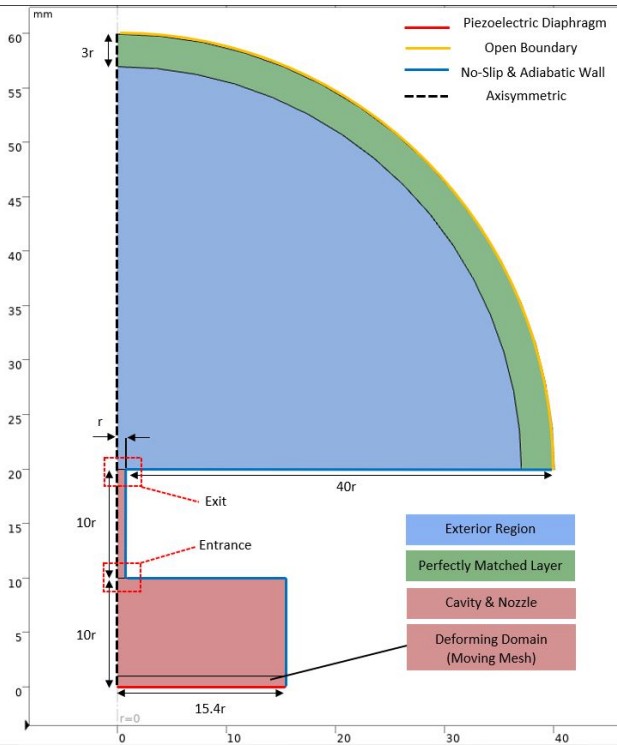

**Figure 2.** Cross-section of cylindrical SJA.

In the experimental setup, sinusoidal voltage signals with excitation frequencies ranging from 100 to 3000 Hz were tested. The voltage amplitude was adjusted to achieve specific cavity pressures, allowing for testing the SJAs under different operating conditions. To assess the performance of the SJAs, the experiments were conducted in quiescent conditions, and flow velocities were measured using hot-wire anemometry. Additionally, the cavity pressure was measured using a microphone. These conditions are matched in the enclosed computational studies.

## 2.1. Diaphragm and Electrostatics

The THUNDER TH-5C piezoelectric actuator diaphragm (manufactured by Face International Corporation) was modeled in this study. It is a composite unimorph ferroelectric actuator consisting of a stainless-steel substrate, a piezoelectric wafer, and an aluminum superstrate [9,32]. These layers are bonded together using an adhesive. The geometry and mechanical specifications of the diaphragm are summarized in Table 1, providing relevant details about its dimensions and characteristics, where $b$, $D$, $E$, and $v$ are the layer thickness, diameter, Young's modulus, and Poisson's ratio, respectively [9,32–36]. The actuator has an overall thickness of $0.48 \pm 0.015$ mm. The stiffness and thickness of the adhesive layers are small relative to the others and were, therefore, excluded from the model.

**Table 1.** THUNDER TH-5C piezoelectric transducer material and geometric properties.

| Layer | Material | $b$ (mm) | $D$ (mm) | $E$ (GPa) | $v$ |
|---|---|---|---|---|---|
| Substrate | Stainless Steel 302 | 0.1524 | 32.77 | 193 | 0.29 |
| Adhesive | LaRC-SI | 0.0381 | 31.75 | 3.8 | 0.4 |
| Piezoelectric | PZT-5A | 0.1905 | 31.75 | 63 | 0.31 |
| Adhesive | LaRC-SI | 0.0381 | 30.73 | 3.8 | 0.4 |
| Superstrate | Aluminum Alloy 2024 | 0.0254 | 30.73 | 69 | 0.33 |

The disk was rigidly attached to its neighboring layers (substrate and superstrate) and clamped around its edge, resulting in the transverse bending of the entire structure when the piezoceramic disk was displaced. Applying an electric potential to the piezoceramic disk elicited a change in shape due to the inverse piezoelectric effect, enabling the SJA to function. Equation (5) is the constitutive relationship for the inverse piezoelectric effect, which relates stress ($\sigma$), strain ($\varepsilon$), and an applied electric field (**E**) in stress-charge form. It can be seen that stress is a function of an applied strain and electric field. This equation is coupled to the classical elasticity Equations (6)–(8), which approximate the deformation of an elastic material under load below its yield stress [27,37]. In these equations, $c_E$ is the elasticity matrix in the presence of a constant electric field, $e^T$ is the piezoelectric constant in the absence of mechanical strain, $\varepsilon_{ij}$ is the strain displacement of the piezoelectric patch, and $u$ is the displacement. Tables 2 and 3 show the relevant material properties for the piezoelectric disk in matrix form [37,38].

$$\sigma = c_E \varepsilon - e^T \mathbf{E} \tag{5}$$

$$\rho \frac{\partial^2 \mathbf{u}}{\partial t^2} = \mathbf{F} \tag{6}$$

$$\mathbf{F} = \nabla \sigma \tag{7}$$

$$\varepsilon_{ij} = \frac{1}{2} \left( \frac{\partial u_j}{\partial x_i} + \frac{\partial u_i}{\partial x_j} \right) \tag{8}$$

**Table 2.** PZT-5A elastic constants, $c_E$ ($\times 10^{10}$ Pa).

| | | | | | |
|---|---|---|---|---|---|
| 12.03 | 7.52 | 7.51 | 0 | 0 | 0 |
| 7.52 | 12.03 | 7.51 | 0 | 0 | 0 |
| 7.51 | 7.51 | 11.09 | 0 | 0 | 0 |
| 0 | 0 | 0 | 2.11 | 0 | 0 |
| 0 | 0 | 0 | 0 | 2.11 | 0 |
| 0 | 0 | 0 | 0 | 0 | 2.26 |

**Table 3.** PZT-5A piezoelectric constants, $e^T$ (C/m$^2$).

| | | | | | |
|---|---|---|---|---|---|
| 0 | 0 | 0 | 0 | 12.29 | 0 |
| 0 | 0 | 0 | 12.29 | 0 | 0 |
| −5.35 | −5.35 | 15.78 | 0 | 0 | 0 |

When a voltage of 424 V (the maximum allowable for this model) was applied to the simply supported actuator in experiments, a maximum center displacement of 0.13 mm occurred [31,34]. To verify the modeling of the THUNDER actuator, a COMSOL multiphysics simulation was performed, coupling solid mechanics and the inverse piezoelectric effect. The substrate, piezoelectric (PZT) layer, and superstrate were modeled with material and geometric properties according to Table 1, with a simply supported boundary condition around the actuator's edge. An electric potential of 424 V was applied to the surface of the piezoelectric layer, and the displacement profile was observed.

From the simulation, the diaphragm maximum deflection was determined to be 0.1302 mm. This maximum displacement represented a 0.2% deviation from the specifications and was, therefore, acceptable for future modeling. A mesh sensitivity study ensured the displacement profile was independent of mesh size (Table 4). While a relatively coarse mesh is suitable to achieve a converged result, 2150 cells were used for subsequent analyses. It should also be noted that 3-D and 2-D axisymmetric models yielded nearly identical results for diaphragm displacement, so 2-D simulations were deemed acceptable.

**Table 4.** Mesh sensitivity analysis of maximum displacement of THUNDER TH-5C actuator in response to 424 V load.

| No. of Cells | 200 | 448 | 1070 | 2150 | 4565 | 9078 |
|---|---|---|---|---|---|---|
| Displacement (mm) | 0.1295 | 0.1297 | 0.1299 | 0.1301 | 0.1302 | 0.1302 |

In the SJA experiment studied by Feero et al. [31], the THUNDER actuator was clamped around its edges. Consequently, in the fully coupled structural–fluidic–acoustic simulation, the boundary condition was adjusted to a clamped configuration, resulting in a significant reduction in the achieved displacement. When clamped, the maximum diaphragm displacement in response to a 424 V input was reduced to 0.0464 mm. The model and relevant boundary conditions are seen in Figure 3, providing an overview of the setup for the diaphragm.

Figure 4 illustrates the radial deformation profile of the top surface of the diaphragm along its radius. The displacement profile exhibits bidirectional behavior, indicating that the same maximum displacement is attained when the diaphragm bends both upward and downward.

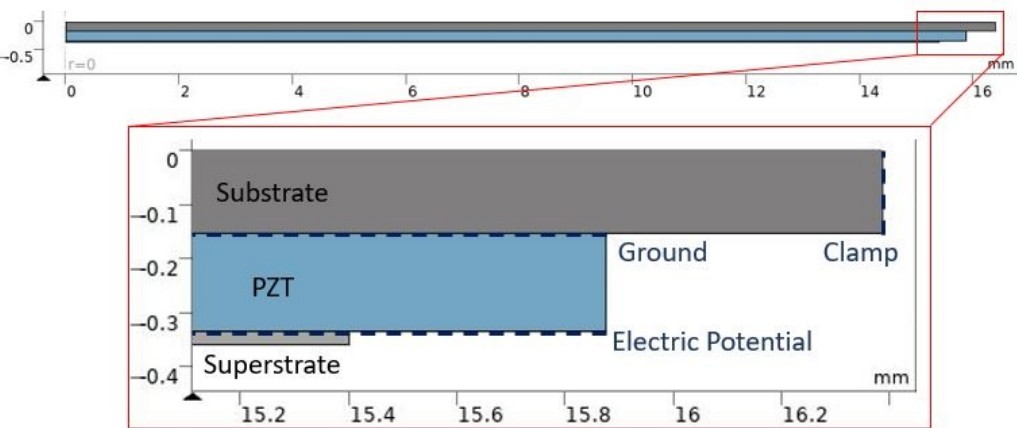

**Figure 3.** Diaphragm boundary conditions.

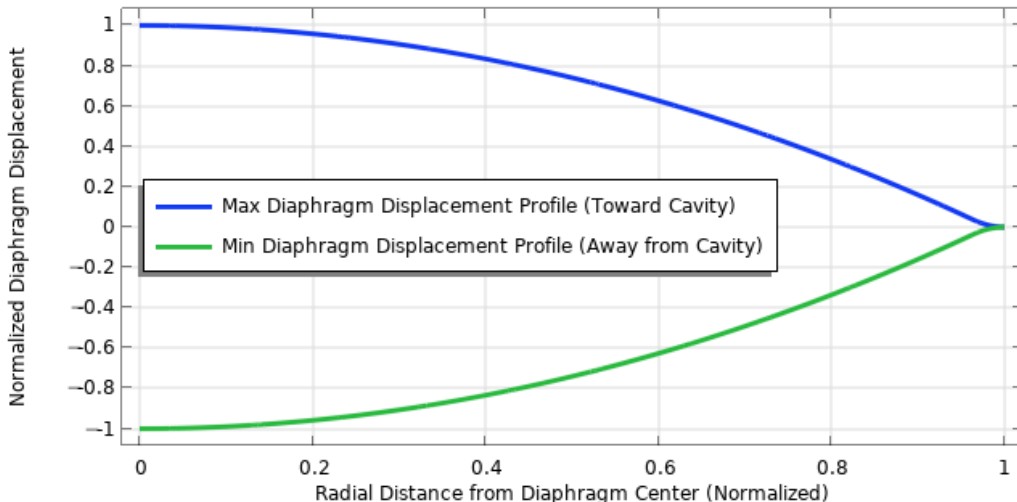

**Figure 4.** Normalized displacement profiles of clamped diaphragm at its extreme positions.

The THUNDER TH-5C actuator has a resonance frequency of 532 Hz when simply supported [34]. An eigenfrequency study was performed using COMSOL to obtain its predicted mechanical resonance frequency. A resonance frequency of 514 Hz was found, which is within 3.4% of the measured value. When perfectly clamped, the predicted mechanical resonance frequency was 2797 Hz, which is 19% higher than the experimentally obtained value of 2350 Hz [31]. This result is in agreement with Gomes [39], who reported that perfect clamping theory tends to over-predict the resonant frequency of piezoelectric diaphragms by 20% because of clamping relaxation (imperfect clamping in experiments) and damping. To account for these factors, the diaphragm was scaled to obtain a resonant frequency that agreed more closely with the experimental results.

To initiate the vibratory behavior of the diaphragm, a time-varying sinusoidal electric potential ($V$) was applied to the diaphragm, where $V_{max}$ is the maximum voltage amplitude (half of the peak-to-peak voltage), $f$ is the excitation frequency, and $t$ is time.

$$V(t) = V_{max} \sin(2\pi f t) \tag{9}$$

The diaphragm excitation frequency was quite large, resulting in the rapid application of the electric potential. This quick voltage application imposed a large gradient, leading to undesirable, transient, high-frequency oscillatory behavior. These oscillations were caused by the diaphragm vibrating at its mechanical resonance frequency in response to the impulse-like application of stress. Over time, these high-frequency oscillations dampened, but they had a negative impact on the initial cycles and the obtained results.

To resolve this issue, a smoothed step function was applied to the electric potential signal. This smoothed step function reduced the initial voltage gradient applied to the diaphragm, eliminating the impulse-like impact of the voltage application. As a result, the diaphragm exhibited a much cleaner displacement profile in response to the voltage application. Moreover, this modification allowed the SJA to reach a steady-state condition in fewer cycles, reducing both computation time and storage requirements. Figures 5 and 6 show how the modified electric potential signal affects the displacement profile of the diaphragm by dampening the impulse-like voltage application.

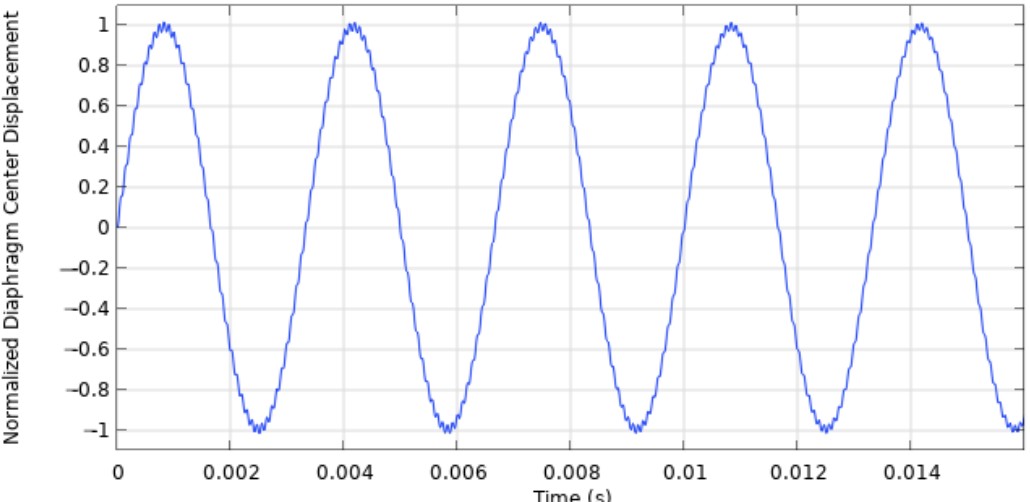

**Figure 5.** Normalized diaphragm center displacement without voltage signal modification.

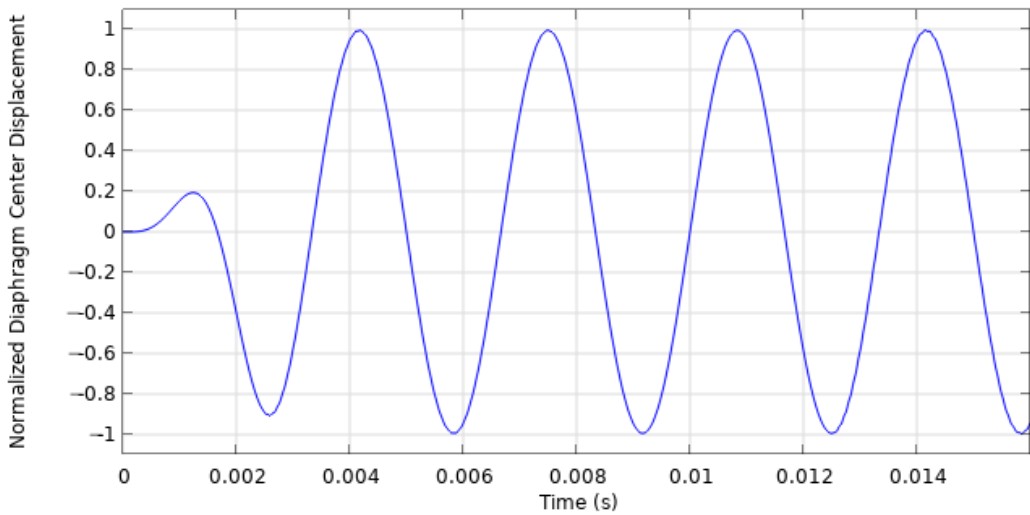

**Figure 6.** Normalized diaphragm center displacement with voltage signal modification.

### 2.2. Pressure Acoustics

Acoustics modeling of the region outside the cavity and nozzle was performed with the *Pressure Acoustics* (COMSOL libraries and modules mentioned in the current work are italicized for emphasis) module along with the *Atmospheric Attenuation* fluid model. This module enabled COMSOL to compute the pressure variations in the fluid to model the propagation of acoustic waves in quiescent conditions. The *Atmospheric Attenuation* fluid model was important, as it is best optimized to describe attenuation due to thermal and viscous effects over long distances and for high-frequency processes. COMSOL computed the acoustic field by solving the governing equations for compressible losses without thermal conduction or viscosity. These equations are the mass, momentum, and

energy conservation equations (Equations (10), (11), and (12), respectively), where $\rho$ is the density of the medium, **u** is the velocity field, $M$ and **F** are possible source terms, and $s$ is entropy [40].

$$\frac{\delta \rho}{\delta t} + \nabla \cdot (\rho \mathbf{u}) = M \tag{10}$$

$$\frac{\delta \mathbf{u}}{\delta t} + (\mathbf{u} \cdot \nabla)\mathbf{u} = -\frac{1}{\rho}\nabla p + \mathbf{F} \tag{11}$$

$$\frac{\delta s}{\delta t} + \nabla \cdot (s\mathbf{u}) = 0 \tag{12}$$

These equations were reduced to an inhomogeneous Helmholtz equation (Equation (13)), where $Q_m$ is the monopole domain source, $q_d$ is the dipole domain source, $c$ is the speed of sound (343 m/s), $k_{eq}$ is the wave number. The $c$ subscript indicates that the variable may be complex-valued [40]. This was achieved by maintaining only the linear terms, assuming that all thermodynamic processes were isentropic, the fluid was stationary, and the pressure field and source terms varied with time in a harmonic manner. Nonlinear effects were ignored because the maximum magnitude of the total acoustic pressure was several orders of magnitude below the threshold of 15 KPa, after which nonlinear effects became relevant [40].

$$\nabla \cdot \left( -\frac{1}{\rho_c}(\nabla p_t - \mathbf{q}_d) \right) - \frac{k_{eq}^2 p_t}{\rho_c} = \mathbf{Q}_m. \tag{13}$$

*2.3. Thermoviscous Acoustics*

Due to the high-frequency oscillation of the pressure field, the viscous and thermal penetration depths (Equations (14) and (15), respectively) had length scales comparable to those of the small geometric features of the nozzle and cavity. Therefore, when modeling the acoustics of the internal regions, the *Thermoviscous Acoustics* module was used. This module computes the acoustic variation of velocity, pressure, and temperature and is required to model acoustics accurately in small-dimensional geometries. For example, when operated at 280 Hz, the viscous and thermal penetration depths near the walls of the SJA in this work were approximately 0.13 and 0.16 mm, respectively. When these penetration depths are comparable to the model geometry, the viscous and thermal losses that occur near walls are significant and must be included explicitly in the equations [40].

$$\delta_v = \sqrt{\frac{\mu}{\pi f \rho_0}} \tag{14}$$

$$\delta_t = \sqrt{\frac{k}{\pi f \rho_0 C_p}} \tag{15}$$

COMSOL simultaneously solved for acoustic velocity variation, acoustic pressure, and acoustic temperature variations using the linearized compressible flow equations (Equations (16)–(21)). Equations (16) and (17) are the Navier-Stokes continuity and momentum equations, respectively, and Equation (18) is the energy equation, where $\Phi$ is the viscous dissipation function, and $Q$ is a heat source. Equation (19) relates the total stress tensor ($\sigma$) and the viscous stress tensor ($\tau$) through Stokes expression, where $\mu_B$ is the bulk viscosity. Equations (20) and (21) are the Fourier heat conduction law and the equation of state, respectively [40]

$$\frac{d\rho}{dt} + \rho(\nabla \cdot \mathbf{u}) = 0 \tag{16}$$

$$\rho \frac{d\mathbf{u}}{dt} = \nabla \cdot \sigma + \mathbf{F} \tag{17}$$

$$\rho C_p \frac{dT}{dt} - \alpha_p T \frac{dp}{dt} = -\nabla \cdot \mathbf{q} + \Phi + Q \tag{18}$$

$$\sigma = -p\mathbf{I} + \tau = -p\mathbf{I} + \mu(\nabla\mathbf{u} + (\nabla\mathbf{u})^T) - (\frac{2\mu}{3} - \mu_B)(\nabla \cdot \mathbf{u})\mathbf{I} \qquad (19)$$

$$\mathbf{q} = -k\nabla T \qquad (20)$$

$$\rho = \rho(p, T) \qquad (21)$$

The boundary between the nozzle exit and the exterior region was an *Acoustic-Thermoviscous Acoustic Boundary*, ensuring there were no discontinuities at the interface between the regions, which were modeled with different physics. A *Thermoviscous Acoustic-Structure Boundary* was used at the interface between the diaphragm and the cavity. This allowed for the two-way coupling of the effect of the diaphragm's solid vibrations on the fluid and the effect of the acoustic pressure waves on the diaphragm.

*2.4. Implementation in the Model*

A frequency domain study was first conducted to investigate the frequencies at which the sound pressure level of the SJA was maximized (indicating optimal performance). This was accomplished by creating a model that included all physics except fluid mechanics and solving for the acoustic field in a stationary domain by exciting the diaphragm over a large range of frequencies. This study was akin to exciting the cavity with a microphone experimentally and measuring the acoustic pressure response to different frequencies. As shown by Figure 7, the results from this study displayed a bimodal peak in the sound pressure level, which is typical of many quantities of interest pertaining to SJAs [23,29–31]. The first peak is associated with the Helmholtz frequency, while the second is associated with the mechanical resonance frequency of the diaphragm. For the purposes of this work and computation of the full model with fluids included, frequencies at or near Helmholtz were considered. In contrast, frequencies near the mechanical resonance were ignored, as jet velocity was maximized near the Helmholtz frequency in [31].

Notably, the Helmholtz frequency predicted by the frequency domain study was 309.5 Hz, which is 0.81% from the theoretical value (307.0 Hz, calculated using Equation (4)) and 10.54% from the experimentally obtained value of 280 Hz from Feero et al. [31]. The difference between the current results and those from the experiments can likely be attributed to small geometric discrepancies, boundary condition idealization within COMSOL (such as sound-hard walls), and measurement techniques. For example, in the experiments, the microphone that measured pressure fluctuations in the cavity was situated behind a pinhole. In contrast, the computational results measured sound pressure levels in the cavity and at the exit of the nozzle.

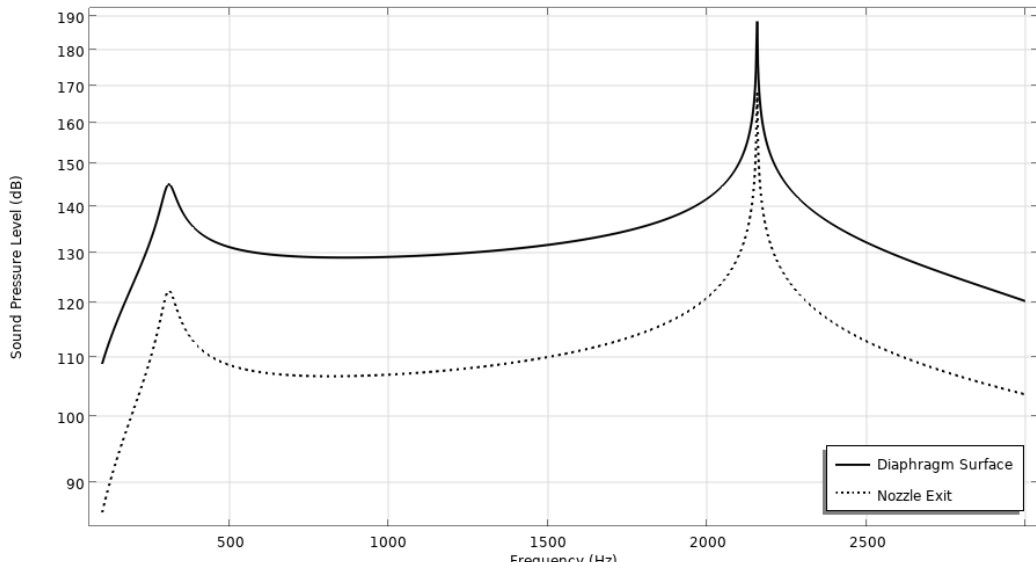

**Figure 7.** Sound pressure level as a function of frequency at the nozzle exit and diaphragm surface.

The model developed for the purpose of this work was transient in nature. Therefore, the transient form of the pressure acoustics and thermoviscous acoustics equations were used. Equation (22) is the scalar wave equation that was used to determine how acoustic waves travel through a fluid in time.

$$\frac{1}{\rho c^2}\frac{\delta^2 p_t}{\delta t^2} + \nabla \cdot \left(-\frac{1}{\rho}(\nabla p_t - \mathbf{q}_{\mathrm{d}})\right) = \mathbf{Q}_{\mathrm{m}} \tag{22}$$

The outer edge of the exterior region of the computational domain was set as a perfectly matched layer. This domain applies complex coordinate scaling to a layer of virtual domains surrounding the physical region of interest. When appropriately tuned, this layer absorbs all outgoing wave energy in frequency-domain problems without any impedance mismatch causing spurious reflections at the boundary [40].

*2.5. Fluids*

The flow field studied in the present work features relatively low fluid velocities and small length scales. Consequently, the jet Reynolds number is well within the laminar range. Feero et al. [31], who studied an SJA with the same characteristics, reported jet Reynolds numbers no higher than 700 and suggested that the jet is laminar in all cases. Therefore, the laminar flow interface was employed within COMSOL.

Gallas [10] showed numerically and experimentally that for excitation frequencies $\frac{f}{f_H} > 0.5$, there is a phase difference between the diaphragm velocity and the nozzle exit velocity, indicating that the flow is compressible. Furthermore, Sharma [23] showed that air in the SJA cavity exhibited compressibility for excitation frequencies above the Helmholtz frequency. Therefore, despite low fluid velocities, compressibility was also captured within the model, as SJAs are typically operated near or above the Helmholtz frequency.

To determine the velocity and pressure of the fluid at each point in the computational domain, the laminar compressible Navier–Stokes equation was used. Equation (23) describes continuity, and Equation (24) describes momentum, where $\rho$ is the fluid density, $\mathbf{u}$ is the fluid velocity vector, $\mu$ is the fluid dynamic viscosity, $\mathbf{I}$ is the identity matrix, and $\mathbf{F}$ shows the body forces on the fluid [41].

$$\frac{\delta\rho}{\delta t} + \nabla \cdot (\rho\mathbf{u}) = 0 \tag{23}$$

$$\rho\frac{\delta\mathbf{u}}{\delta t} + \rho\mathbf{u} \cdot \nabla\mathbf{u} = -\nabla p + \nabla \cdot \left(\mu(\nabla\mathbf{u} + (\nabla\mathbf{u})^T) - \frac{2}{3}\mu(\nabla \cdot \mathbf{u})\mathbf{I}\right) + \mathbf{F} \tag{24}$$

The boundary conditions for the fluid mechanics of the model in this work can be seen in Figure 2. All walls had no-slip boundaries. The region outside the SJA was taken as a quiescent flow in atmospheric conditions. Therefore, *Open Boundary* was used to simulate the limit between the computational domain and the rest of the same fluid not represented in the geometry. It was characterized by a normal stress of zero [41],

$$f_0 = 0 \tag{25}$$

The fluidic–structural relationship was fully coupled with COMSOL's *Fluid-Structure Interaction* module, meaning the interaction of the structure on the fluid, as well as the fluid on the structure, was modeled. Though the diaphragm deflects minimally, its variations in time were rapid, causing pressure waves to radiate in the fluid, meaning this interaction was important to capture properly. The motion of the fluid also imparted its inertia on the diaphragm surface. Coupling this interaction also ensured the fluid in the cavity dampened the motion of the diaphragm. Small-scale oscillations present in the diaphragm's transient response were quickly dampened by the inertia of the fluid it displaced. The fluid pressure also impacted the diaphragm motion by reducing the expected maximum inward displacement and increasing the outward deflection (away from the cavity), meaning the

transient behavior of the diaphragm when operated in the SJA was not bi-directional, as it is when freely vibrating. Because of this phenomenon, the diaphragm deflected approximately 10% further away from the cavity rather than toward it. The total force exerted on the solid boundary from the fluid (the negative of the reaction force on the fluid) is described by Equation (26), where $p$ denotes fluid pressure, and Equation (27) describes the motion of the diaphragm, which acts as a moving wall for the fluid domain, where **n** is the outward normal to the diaphragm boundary.

$$\mathbf{f} = \mathbf{n} \cdot \left\{ -p\mathbf{I} + \left( \mu(\nabla\mathbf{u}_{\text{fluid}} + (\nabla\mathbf{u}_{\text{fluid}})^T) - \frac{2}{3}\mu(\nabla\cdot\mathbf{u}_{\text{fluid}})\mathbf{I} \right) \right\} \tag{26}$$

$$\frac{\delta\mathbf{u}_{\text{struct}}}{\delta t} \tag{27}$$

Both 2-D axisymmetric and 3-D simulations were performed, and their results were compared. The primary performance variable for SJAs (nozzle exit centreline velocity) was taken as the comparison metric. Maximum nozzle exit velocities for four different excitation frequencies were compared and are shown in Table 5. It can be seen that the predicted maximum nozzle exit velocities were comparable to within 2.0%, indicating that a 2-D simulation was adequate for the purposes of this study. Therefore, the majority of the results in this work were obtained in 2-D axisymmetry, where the vertical centerline is given an axisymmetry boundary condition.

**Table 5.** Predicted Maximum Nozzle Centreline Exit Velocity for 2-D and 3-D Models.

| Excitation Frequency (Hz) | 270 | 285 | 300 | 310 |
|---|---|---|---|---|
| Max. Centerline Velocity: 2-D (m/s) | 3.98 | 4.33 | 4.42 | 3.80 |
| Max. Centerline Velocity: 3-D (m/s) | 4.02 | 4.39 | 4.49 | 3.85 |

*2.6. Computational Domain and Meshing*

To ensure that the mesh refinement had negligible influence on the results of the simulations, a mesh sensitivity study was performed following the methods outlined by Celik et al. [42] based on the nozzle exit velocity [20,22,27]. It was found that the chosen mesh (with 40,577 elements) was adequate.

The computational domain for the simulations comprised three subdomains. The first subdomain was the cavity and nozzle. This region was meshed with a fine triangular mesh, a boundary layer on the walls, and extra refinement at the entrance and exit of the nozzle to accommodate the high gradients near the corners. Notably, Jain et al. [20] reported that grid refinement at the nozzle exit was most crucial, as the results are most sensitive in that region, and stated that a resolution of 0.1 mm was sufficient for an orifice diameter of 3 mm. Taking this into account, the selected mesh for the current research had 14 elements across its 1 mm radius, which resulted in an average resolution of 0.07 mm at the exit.

Given the dependence of thermal and viscous penetration depths on frequency, the boundary layers were given a thickness according to Equation (15), which varied based on the actuation frequency of the diaphragm. For example, the boundary layers were 0.16 mm thick when the actuation frequency was 280 Hz. All boundary layers were comprised of 8 layers. A region was also implemented near the diaphragm for moving mesh to accommodate the motion of the diaphragm. Yeoh smoothing was used for mesh smoothing [43].

The exterior region of the computational domain comprised fine triangular elements near the nozzle exit, which became coarser as the distance from the nozzle grew. The maximum element size in the near-orifice region was set to 1/3 of the radius of the nozzle, as recommended by Jensen [44] when modeling thermoviscous acoustics. This was also sufficient to capture the vortex roll-up and high-velocity gradients. The mesh was coarsened beyond 12 orifice diameters from the nozzle exit. The radius of the exterior region was set

to 20 times that of the orifice radius because Qayoum and Malik [27] showed that it was sufficient to ensure the boundary had a negligible effect on results.

The final region of the computational domain was the perfectly matched layer that surrounded the exterior region. This region was meshed with quadrilateral elements and was 8 layers thick [40]. The entire mesh consisted of 40,577 elements with an average quality of 0.87. The mesh used by Qayoum and Malik [27], who also studied SJAs from a multiphysics perspective and at comparable Reynolds numbers, was used as a reference. The number of elements in the nozzle and cavity was based loosely on the findings of their mesh sensitivity study. Quadratic elements were used to compute electrostatics, solid mechanics, and pressure acoustics, as their governing PDEs have dominant second derivatives. Lagrange and serendipity shape functions were also employed for solid mechanics and pressure acoustics, respectively. Linear elements with Lagrange shape functions were used to calculate fluid mechanics to maintain computational efficiency and achieve satisfactory accuracy.

The region outside the cavity and nozzle was modeled as a hemisphere because the sound emanates from the nozzle exit, which can be approximated as a sound point source. This is favorable for the setup of the perfectly matched layer. Figure 8 shows the mesh that was selected for the simulations. Figures 9 and 10 show the mesh refinement at the nozzle exit and entrance, respectively.

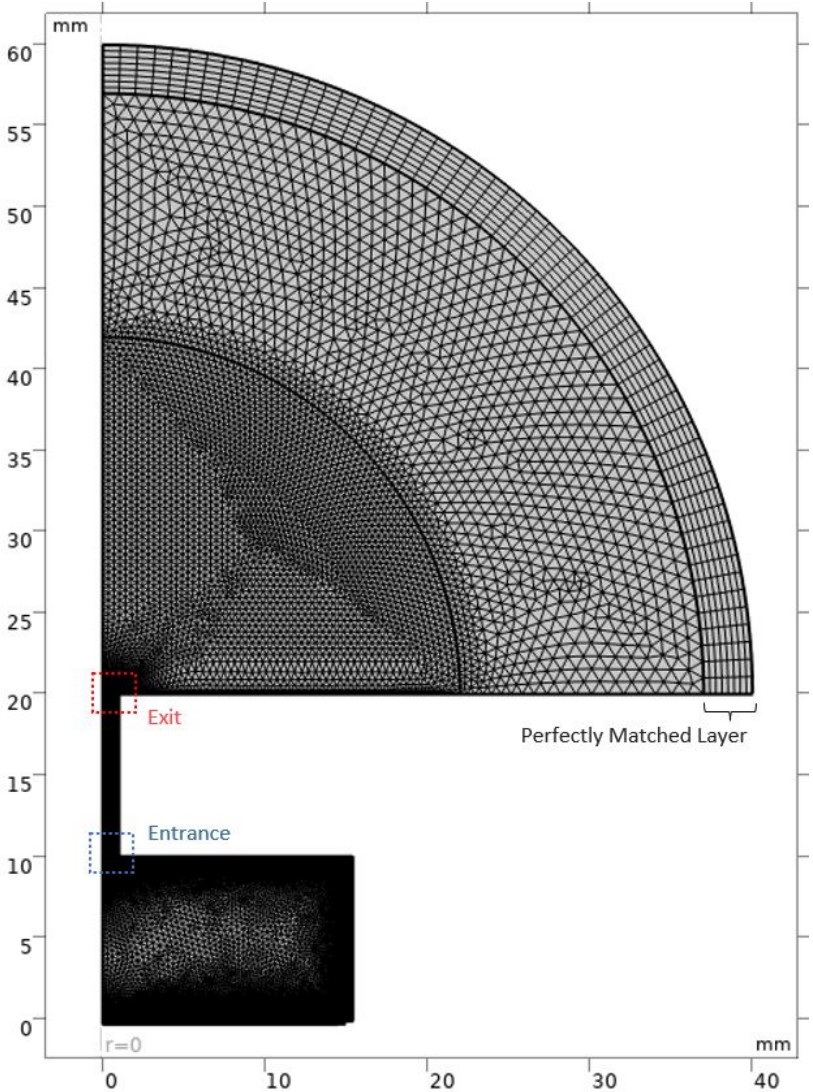

**Figure 8.** 2-D axisymmetric mesh with important regions highlighted.

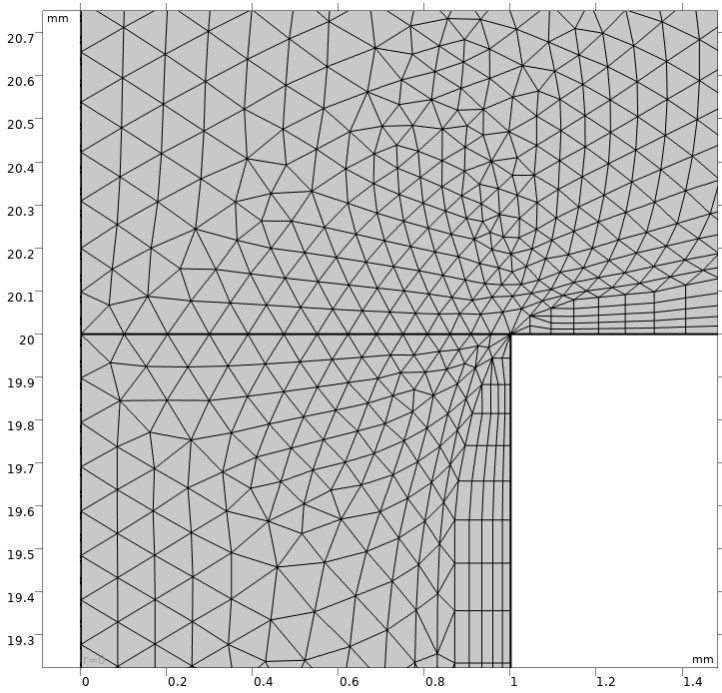

**Figure 9.** Mesh at nozzle exit.

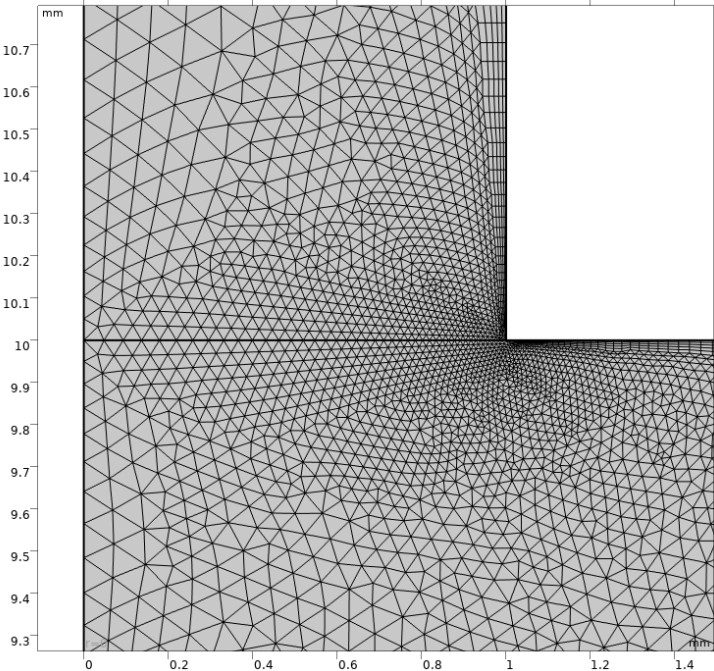

**Figure 10.** Mesh at nozzle entrance.

### 2.7. Time and Solver Settings

Synthetic jet actuators have flow fields that are transient in nature. Therefore, an unsteady study was performed. The first cycles of the transient study establish a steady-state operation condition of the SJA, after which each cycle of operation performs virtually the same as the last. To ensure that the results were independent of the total discretization time of the simulation, each trial was performed for a minimum of 15 cycles. Figure 11 shows the centerline velocity at the nozzle exit of the SJA for the first 15 cycles. After the 10th cycle, the maximum velocity in each cycle varies by no more than 0.1 m/s.

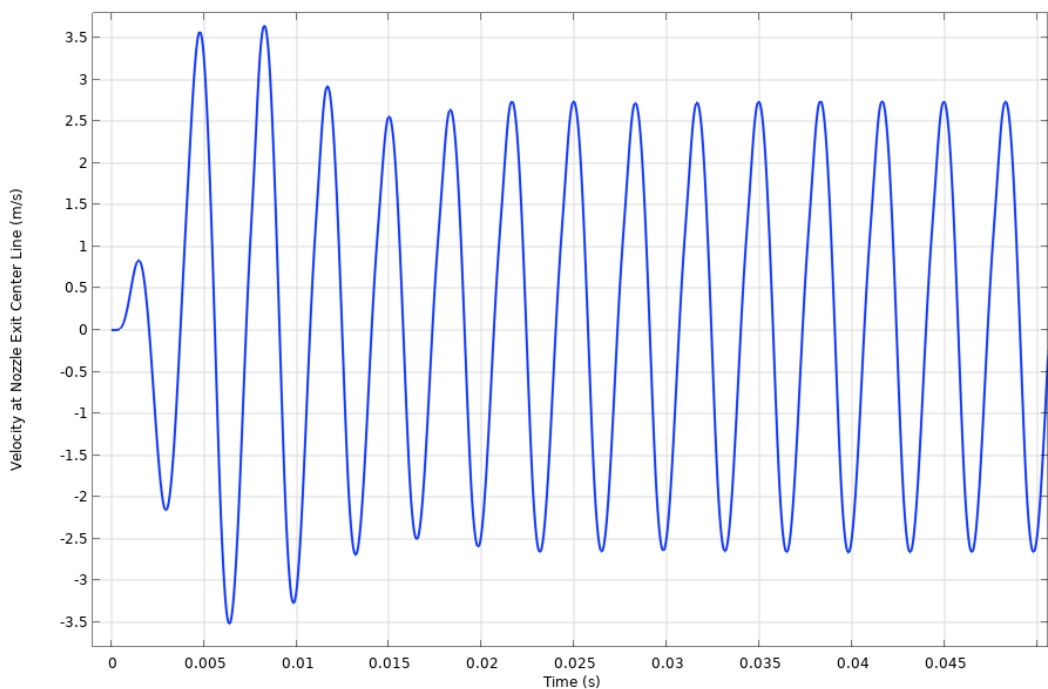

**Figure 11.** SJA nozzle exit centerline reaching steady state after a transient start-up condition.

Adaptive time stepping allowed the solver to use the largest possible time step for each iteration while maintaining accuracy. An upper bound for time step size of $1/(1000f)$ was implemented to ensure there were enough data points at the peaks. This was inspired by Jain et al. [20] and Qayoum and Malik [27], who found this time step size to capture both positive and negative peaks adequately. However, Qayoum and Malik [27] also claimed that velocity is insensitive to the time step. Simulations were considered converged when relative residuals of all relevant variables reached $10^{-5}$ and the peak centerline exit velocity varied by less than 1% from the previous cycle. Figure 12 shows a section of the convergence plot generated by COMSOL during computation. A folder was created to store all the simulation information and results so that future work can use it as a starting point.

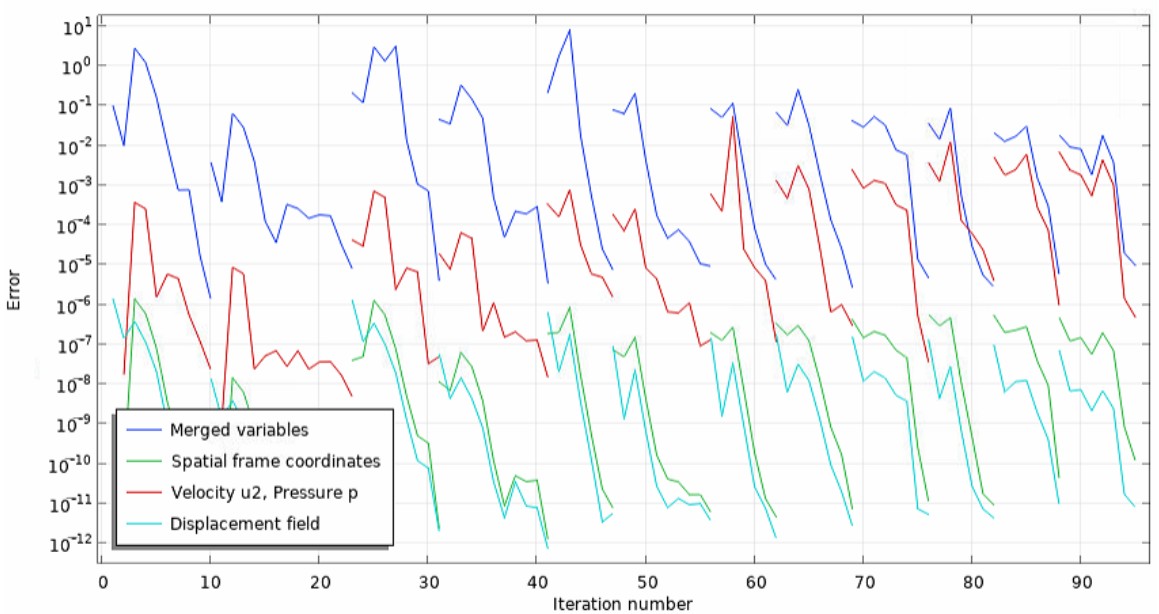

**Figure 12.** Section of convergence plot during computation.

## 3. Results

### 3.1. Fluids

The synthetic jet actuator frequency response was established by determining the maximum exit velocity of the jet at its centerline for frequencies from 100 to 1000 Hz. To do this, a frequency sweep with 10 Hz increments near the Helmholtz frequency and 50–100 Hz away from it was performed. Figure 13 shows the frequency response of the SJA used in the current work. Frequencies near the mechanical resonance frequency of the diaphragm (2350 Hz) were not investigated, as the hot-wire experiments found no significant peak in that range. This is likely due to the geometry of the SJA. Many studies have utilized SJA cavities with "pancake" shapes, meaning they are short with an aspect ratio on the order of 10 or higher [19,27–30]. These configurations yield significant peaks in jet velocity near the mechanical resonance frequency of the diaphragm because the volume of fluid displaced by the diaphragm is significant compared to the volume of the cavity. The current study considers an SJA with an aspect ratio of $d_{cavity}/h_{cavity} = 3.08$.

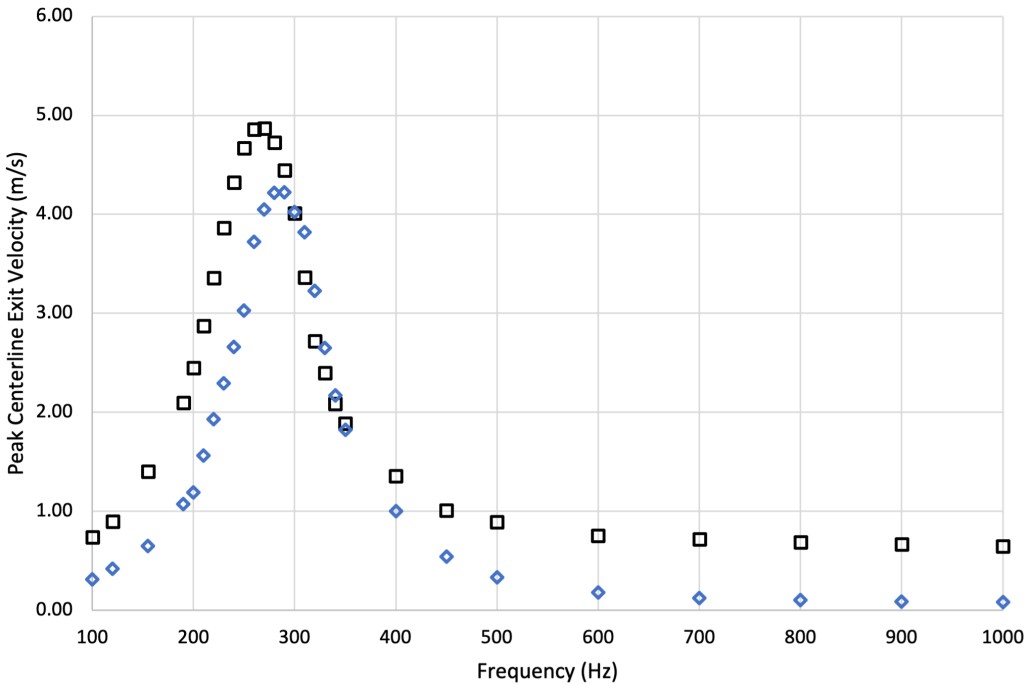

**Figure 13.** Synthetic jet actuator centerline exit velocity response. Markers indicate experimental hot-wire measurements [31] (blue ◇) and the current multiphysics model (black ☐).

Figure 13 also includes the hot-wire velocity measurements from Feero et al. [31]. It can be seen that the current work captured the location of the velocity peak associated with the Helmholtz frequency closely. The current (multiphysics) work predicted that the SJA performance (center line velocity) is maximized when the actuation frequency is 267.1 Hz. This peak, associated with the Helmholtz frequency, is 12.9% from the theoretical Helmholtz frequency (307.0 Hz), 4.7% from the experimentally determined Helmholtz frequency (280 Hz), and 6.6% from the velocity peak in the hot-wire experiments (286.0 Hz) [31]. The magnitude of the maximum centerline velocity predicted by the multiphysics model showed reasonable agreement with the experimental results. The multiphysics model predicted the highest centerline velocity of 4.87 m/s, which is an over-prediction of 15.4%. The centerline velocity hot-wire measurements were taken 0.2 mm downstream of the orifice, whereas the computational results were taken at the exit plane of the orifice. Furthermore, hot-wire measurements cannot capture flow direction, and the computational model assumes perfect electrostatic conditions (no power loss). All of these factors may

play a role in the over-prediction of the maximum jet velocity. These results are summarized in Table 6.

**Table 6.** Synthetic jet actuator characterization: location of centerline velocity peak associated with Helmholtz frequency.

|  | Theoretical | Model | Experiment |
| --- | --- | --- | --- |
| Frequency (Hz) | 307.0 | 267.1 | 286.0 |
| Jet Velocity (m/s) | N/A | 5.05 | 4.25 |

Another parameter of interest when analyzing synthetic jet actuators is the radial profile of axial jet velocity. During expulsion, a "top-hat" velocity profile is the optimal case. This profile indicates that velocity is constant across the width of the orifice [9]. Feero et al. [31] performed hot-wire measurements across the diameter (from $r/d = -0.5$ to 0.5) to measure this radial profile for different phases of operation of the SJA. Ziade et al. [22] performed a numerical study with the same geometry using a second-order scheme and also plotted their radial profiles of axial jet velocity. The results of Feero et al., Ziade et al., and the current work can be seen in Figure 14. The diaphragm actuation frequency was 300 Hz. The velocities are normalized by their respective maximum centerline jet velocity. It should be noted that there is significant uncertainty in the hot-wire measurements near the orifice edges. Again, some minor discrepancies exist because of measurement techniques. The hot-wire measurements were taken 0.15 mm downstream of the orifice, but the current work used the orifice exit plane to record the velocity profiles.

The overall agreement between the current model and the experiments is excellent. The multiphysics model was able to capture the shapes of all profiles accurately despite the velocity profile during expulsion, ($0° < \phi < 135°$) (Phase angle $\phi$ indicates at what point in a cycle (one period) a measurement or computation was made. An angle of $\phi = 90^o$ corresponds to the maximum expulsion velocity.), deviating significantly from the ideal "top-hat" shape. The velocity profile during the first half of expulsion is representative of fully developed flow in a circular duct, which is driven by an oscillating pressure gradient [31,45]. Fully developed flow is expected in this case, as the nozzle length-to-diameter ratio is relatively large. This type of flow is characterized by the Stokes number ($S$), which is approximately 22 for this case. White [45] also states that as $S$ reaches about 20, the average velocity reaches 90% of the centerline velocity and approaches 100% as $S$ approaches infinity [31]. The ingestion portion of the cycle also shows excellent agreement with the hot-wire measurements and other computational work. The flow during ingestion is a more typical "top-hat" shape, as expected.

Figures 15 and 16 show the velocity and vorticity magnitude contours, respectively, of the SJA at six different phase angles in one cycle. The contours are from the 20th cycle in a simulation of the SJA operating at an actuation frequency of 300 Hz.

The beginning of expulsion ($\phi = 45°$) shows the early stage of the formation of a vortex ring at the orifice exit. The fluid is just exiting the orifice and has begun to curl over the edges, creating an area of circulation, as shown by the vorticity contours (Figure 16a). During this time, vortices inside the cavity from the previous cycle have detached from the nozzle entrance and are traveling back toward the diaphragm, and vortices outside the cavity permeate downstream. The vortex from two cycles prior has impacted the diaphragm and has begun to dissipate along its surface. The development of the vortex ring is depicted nicely during the next phases of expulsion (Figure 16b,c). These contours show a clear picture of "vortex roll-up". The velocity in the nozzle also begins to decrease as the ingestion phase is set to begin shortly. As ingestion begins ($\phi = 225°$), the vortex ring outside the orifice forms fully as fluid rushes into the orifice behind it. The vorticity contour (Figure 16d) shows that a coherent structure has formed, proving that the vortex detaches and is not ingested back into the cavity. Simultaneously, the formation of a vortex ring begins inside the cavity. Recirculation zones are noted at the nozzle exit, where fluid

travels around the orifice edge. The rest of the ingestion cycle ($\phi = 270°, 315°$) sees the completion of the vortex ring inside the cavity and the arrival of the previous vortex ring at the diaphragm. Figure 15e,f show the velocity contours of fluid that was ejected from the cavity during this cycle and now travels downstream. This summary explains qualitatively how a synthetic jet actuator works, with imagery to illustrate its inner workings.

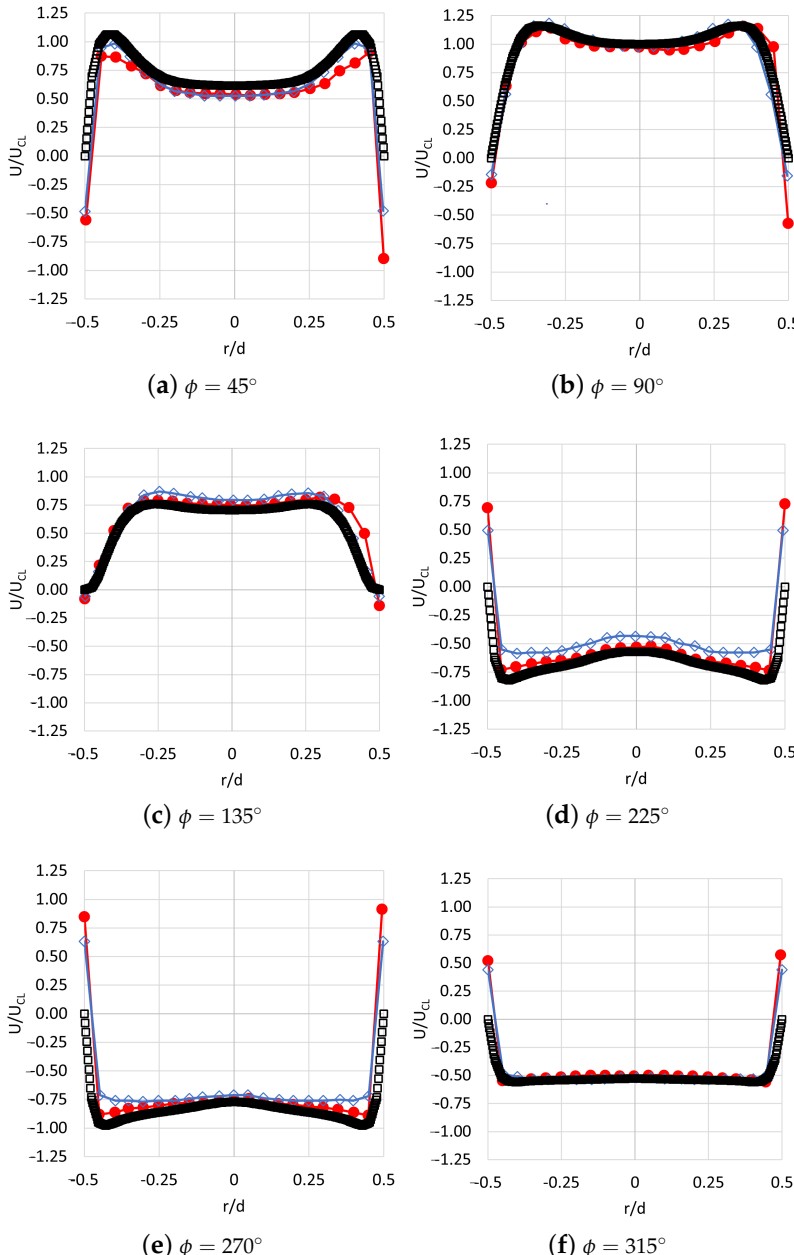

**Figure 14.** Radial jet velocity profiles at SJA nozzle exit, normalized by maximum centerline velocity. Markers indicate hot-wire [31] (red ○), numerical [22] (blue ◇), and multiphysics (black □) results.

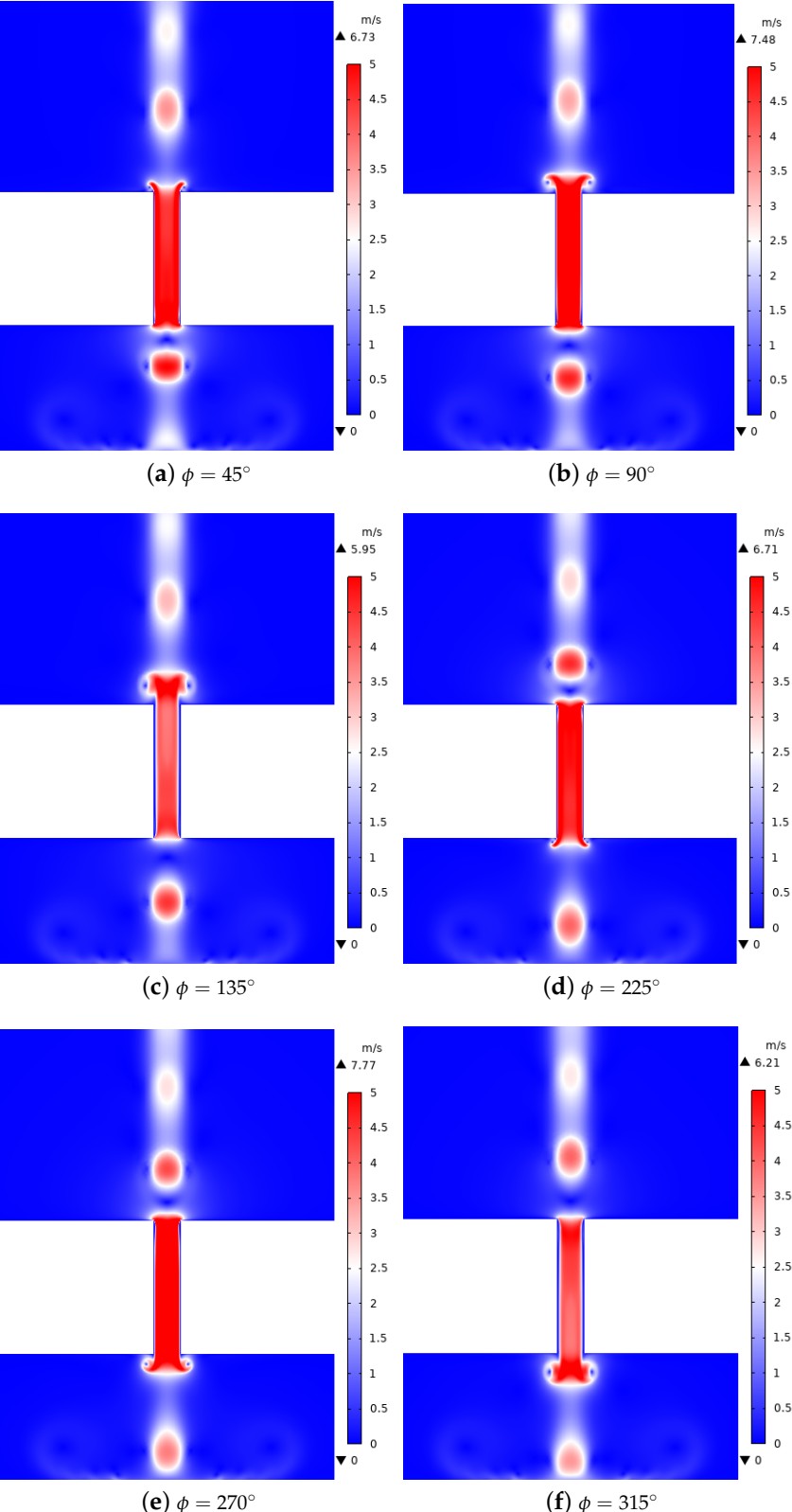

**Figure 15.** Velocity magnitude contours for one cycle.

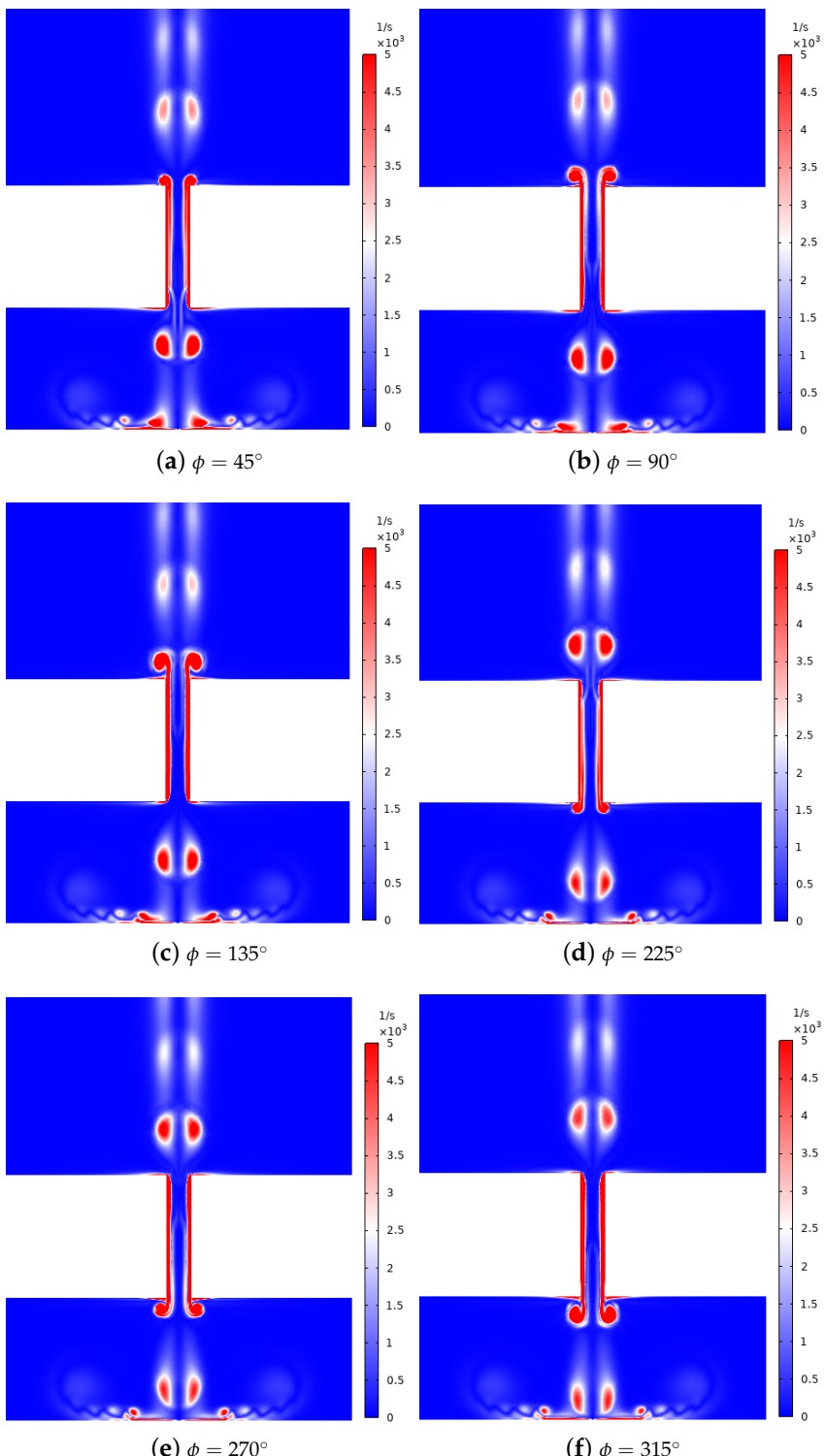

**Figure 16.** Vorticity magnitude contours for one cycle of the multiphysics model.

Van Buren et al. [46] found that a phase shift between the diaphragm displacement and jet velocity exists, indicating that the flow is compressible within the cavity and/or nozzle. This phenomenon occurs as a product of solving the Navier–Stokes equations for channel flow with an oscillating pressure gradient. Despite low fluid velocities relative to supersonic flows, several studies have reported that compressibility should be considered when modeling synthetic jet actuators. As indicated earlier, Gallas [10] stated that compressibility effects must be considered when the actuation frequency is $f \geq 0.5f_H$.

Sharma [23] reported that a phase shift (and, thus, flow compressibility) began to form near the Helmholtz frequency and grew until the diaphragm displacement and orifice velocity were completely out of phase. As the current work focuses on actuation frequencies near the Helmholtz frequency, compressible flow was used. Figure 17 shows the oscillations of the diaphragm velocity and displacement at its center and jet velocity at the exit of the nozzle in response to a 300 Hz actuation frequency. Clearly, a phase shift exists between the variables, indicating that compressibility must be accounted for to accurately model SJAs. At an actuation frequency of 300 Hz, a phase shift between the diaphragm displacement and jet velocity is approximately 51°.

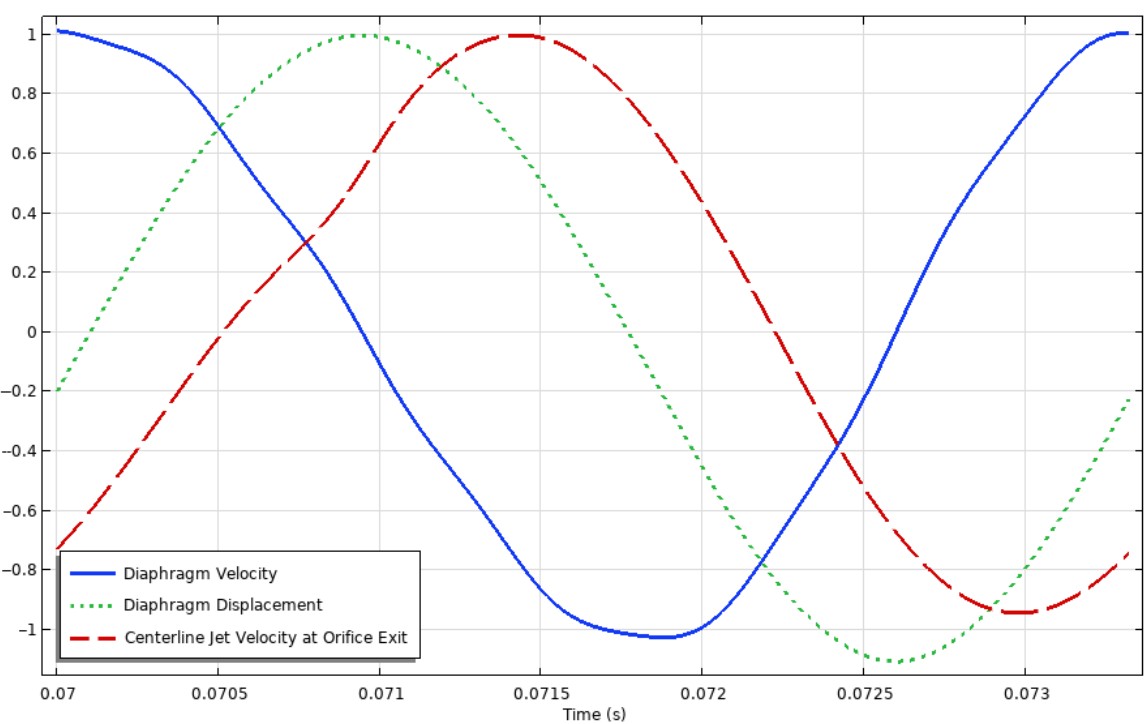

**Figure 17.** Normalized diaphragm and flow variables in response to a 300 Hz actuation frequency.

When compressibility is introduced, temperature effects become relevant as well. Van Buren et al. [46] discussed this briefly in their work on high-speed and momentum SJAs, where they found that an increase in temperature was in the nozzle due to fluid compressibility. Temperature variation inside the nozzle was seen in the current work but is negligible because fluid velocities are relatively low.

### 3.2. Acoustics

As with fluid behavior, the noise generated by synthetic jet actuators is highly frequency-dependent. The noise generated by SJAs is maximized when the actuation frequency matches the Helmholtz frequency or the mechanical resonance of the diaphragm. Arafa et al. [6] showed that at higher harmonics, noise generation could be lower at a minimal cost to the jet's performance. For the purposes of the current work, however, only frequencies near the Helmholtz frequency were considered. Figure 18 shows the maximum centerline acoustic pressure at the orifice exit and at the center of the diaphragm's surface. The noise generation for both locations was maximized at the Helmholtz frequency, as expected. Interestingly, the position of the peak acoustic pressure is at 310 Hz, which is only 0.98% from the theoretical value of 307.0 Hz. This was the expected behavior of the jet velocity, which peaked at an actuation frequency of 267.4 Hz.

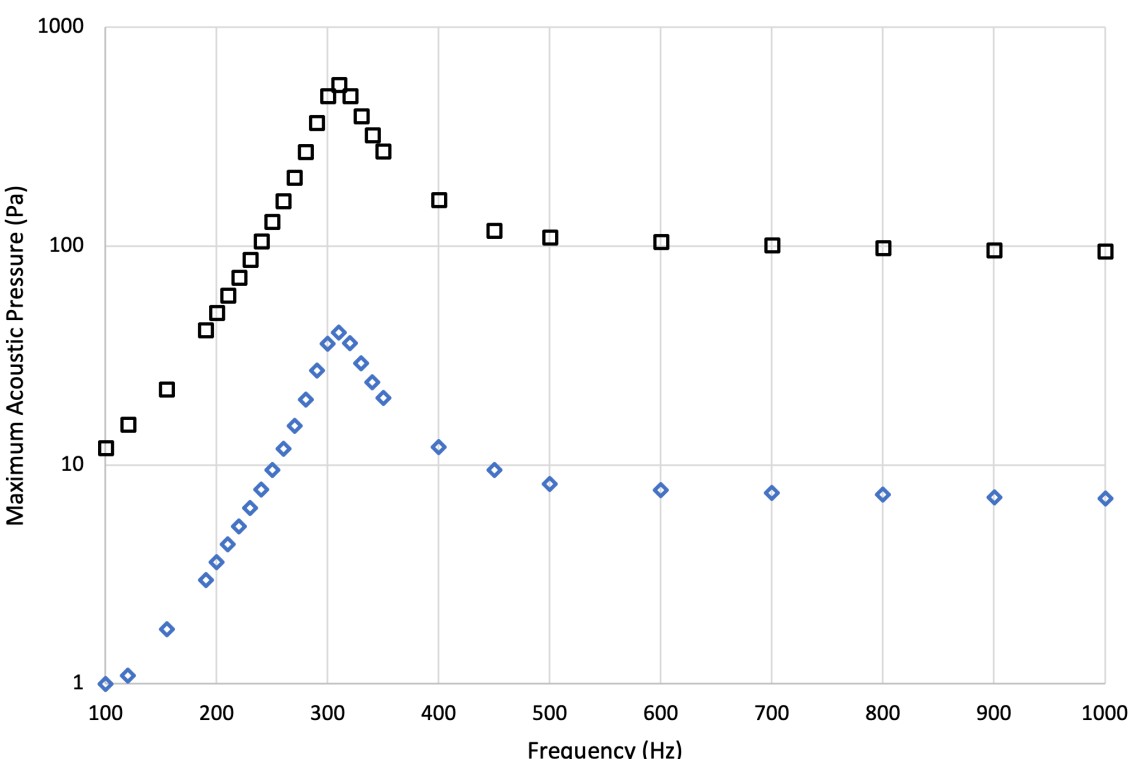

**Figure 18.** Maximum acoustic pressure at the orifice exit as a function of actuation frequency. Markers indicate orifice exit centerline (blue ◇) and diaphragm surface (black □).

The form of the acoustic signal is also important to understand. Mallionson et al. [47] reported that the mechanical resonance of the diaphragm and the acoustic resonance of the cavity influence the jet plume formed by SJAs. Furthermore, numerical work by Wang et al. [3] found that the sound generated by SJAs can be broken down into two monopole components. These monopoles are structure-borne noise (caused by the flexing of the diaphragm and its interaction with the fluid) and fluid-borne noise (related to the resonant frequency of the cavity). They found that cancellation exists between these monopoles at low frequencies when their signals are out of phase.

When modeling pressure acoustics without thermoviscous effects, the viscous- and thermal-related losses are not accounted for. Initial trials for the current work modeled the acoustics of SJAs with linear elastic behavior (lossless). While this is not physically realistic, it does present an interesting discussion. When the viscous and thermal losses are ignored, the transient acoustic response develops a beat frequency—a consequence of the cancellation of the structure-borne and fluid-borne sound monopoles. Figures 19 and 20 show the transient acoustic response from exciting the synthetic jet actuator at 250 *Hz* and 320 Hz, respectively, in the absence of thermoviscous losses. This is a beat frequency and a result of alternate addition and cancellation of sources that are out of phase. In this case, 250 Hz is approximately 57 Hz from the Helmholtz frequency of the cavity. Therefore, the two sound sources (diaphragm and cavity) interfere and partially cancel each other approximately every 1/57 = 0.0175 s. Similarly, 320 Hz is approximately 13 Hz from the Helmholtz frequency, meaning the signals cancel each other approximately every 1/13 = 0.077 s. This is illustrated further by Figures 21 and 22, which are discrete Fourier transforms of the transient acoustic signals. Figure 21 has two large distinct peaks, corresponding to the actuation frequency (250 Hz) and the Helmholtz frequency (307.0 Hz). Conversely, Figure 22 has only one distinct peak. This is because the figure's sensitivity is too low to display the actuation and Helmholtz frequencies separately. Furthermore, because these peaks are much closer together, they superimpose to generate more noise.

The preceding peaks shown in the Fourier transforms correspond to twice the actuation frequency ($2f$) and the mechanical resonance frequency of the diaphragm.

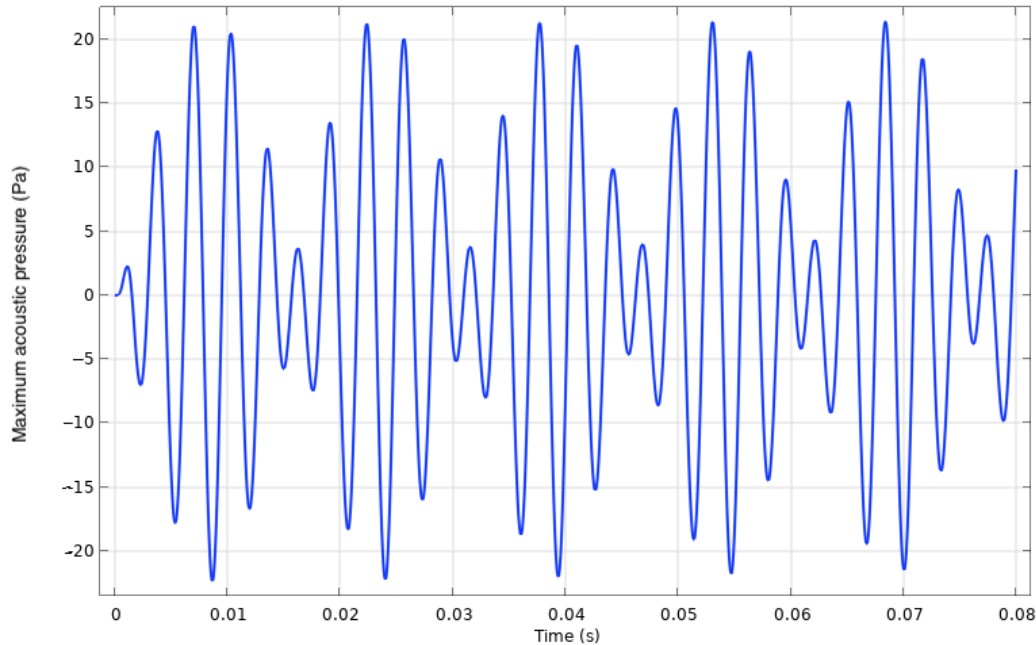

**Figure 19.** Acoustic transient response of synthetic jet actuator at its orifice operating at 250 Hz without thermoviscous losses.

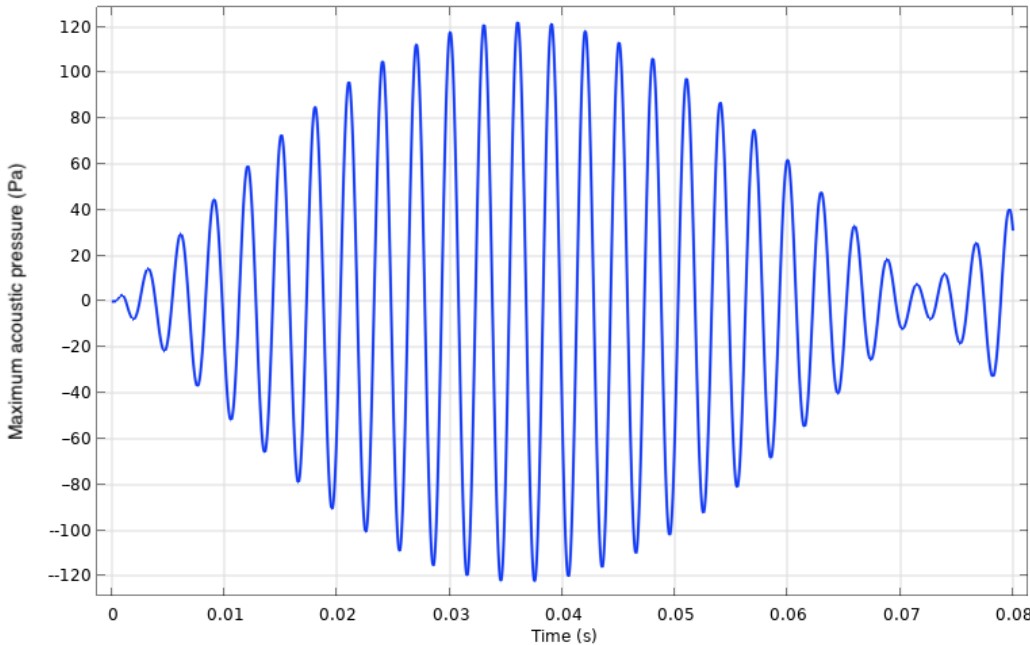

**Figure 20.** Acoustic transient response of synthetic jet actuator at its orifice operating at 320 Hz without thermoviscous losses.

These findings, while interesting, ignore thermoviscous losses that take place in the nozzle and cavity. When thermoviscous losses were accounted for, the acoustic behavior did not display the same beat frequency. Figures 23 and 24 show the transient response and discrete Fourier transform of the response, respectively, when the diaphragm is excited 320 Hz and thermoviscous losses are incorporated. In the transient response, it is evident that the beat frequency behavior has disappeared. The transient response reaches a steady state condition after about 12 cycles. The Fourier transform supports this, as the

Helmholtz frequency peak has been almost entirely damped out. The only prominent peak is associated with the actuation frequency.

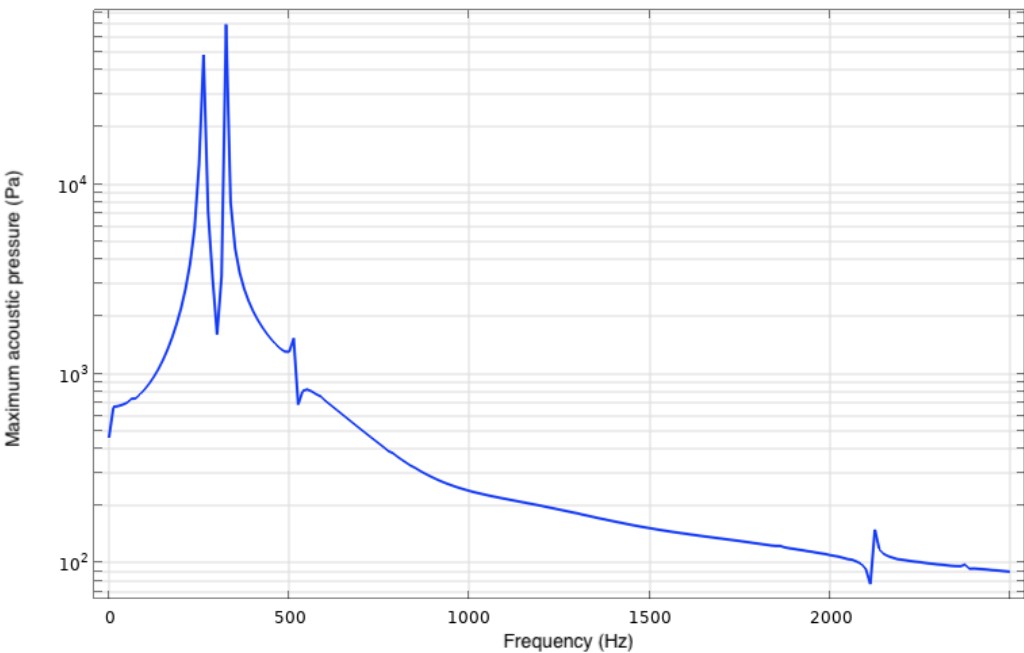

**Figure 21.** Discrete Fourier transform of transient acoustic response of synthetic jet actuator operating at 250 Hz without thermoviscous losses.

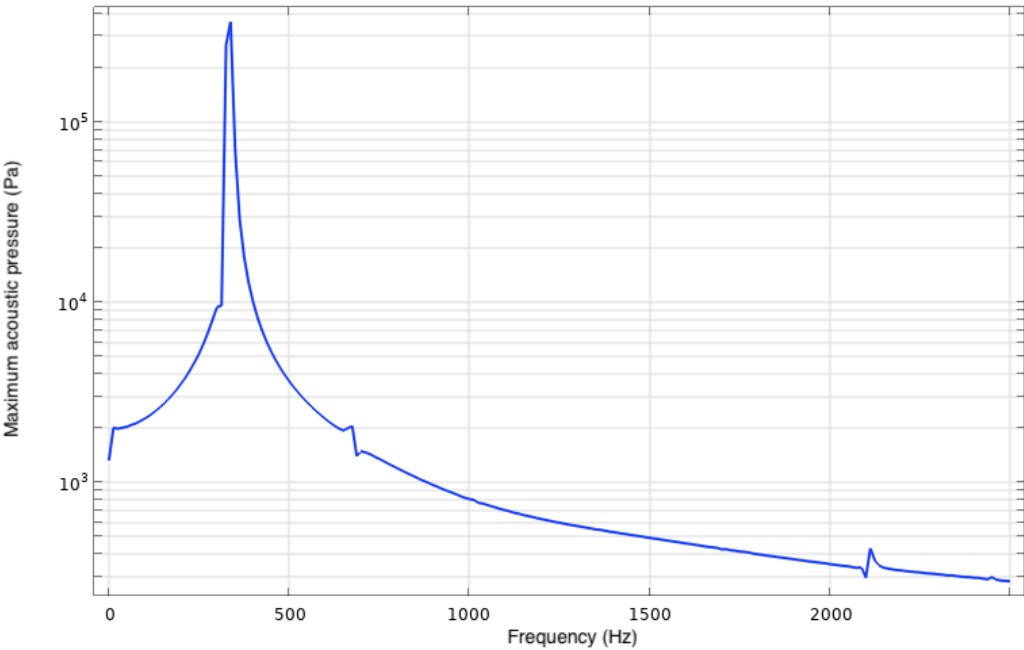

**Figure 22.** Discrete Fourier transform of transient acoustic response of synthetic jet actuator operating at 320 Hz without thermoviscous losses.

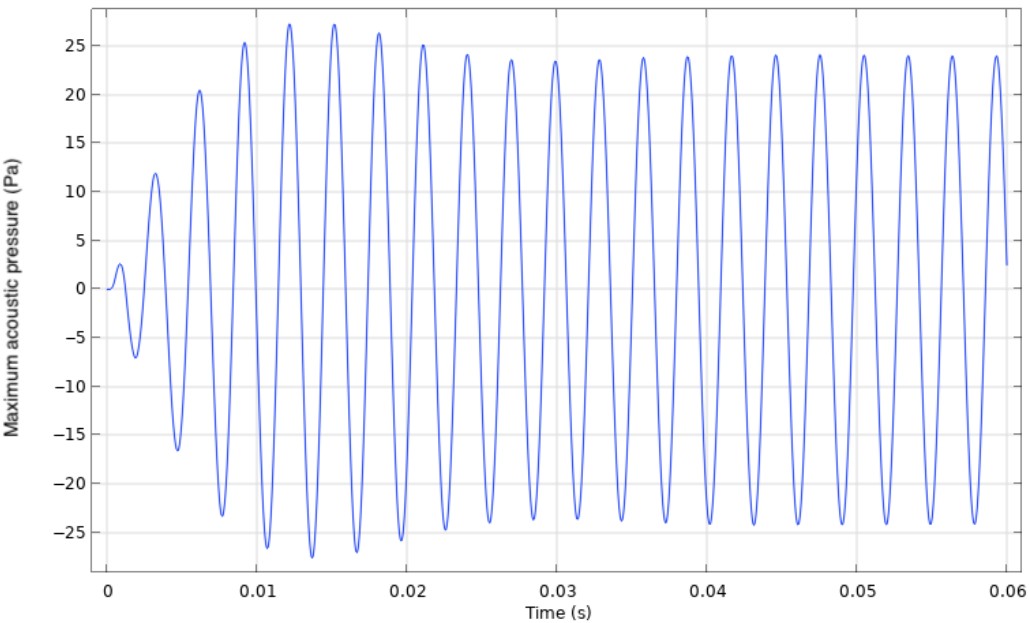

**Figure 23.** Transient acoustic response of synthetic jet actuator operating at 320 Hz with thermoviscous losses.

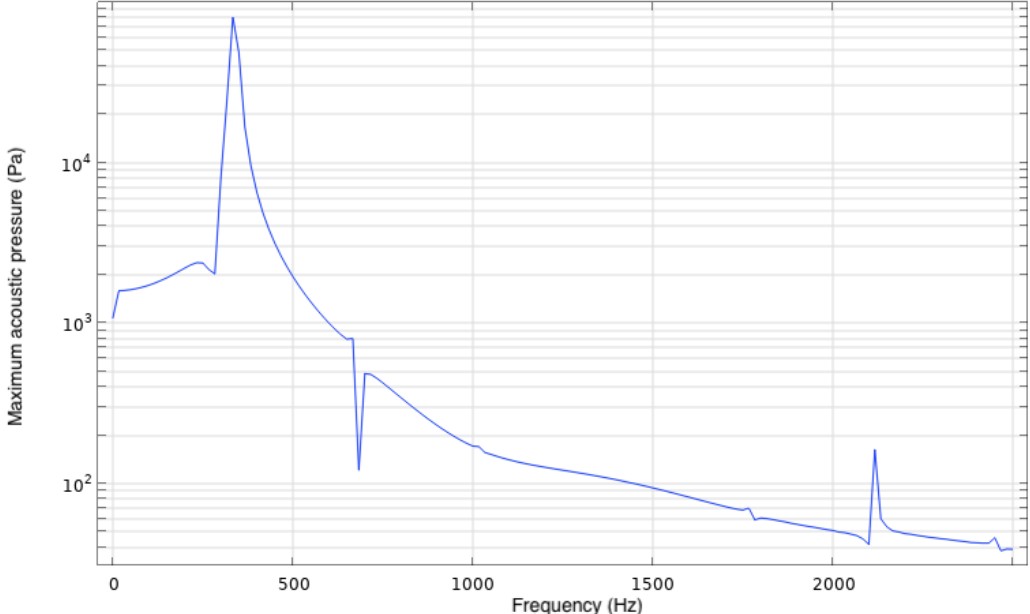

**Figure 24.** Discrete Fourier transform of transient acoustic response of synthetic jet actuator operating at 320 Hz with thermoviscous losses.

The same conclusions can be drawn regarding the SJA driven with an actuation frequency of 250 Hz (Figures 25 and 26). The only prominent peak that remains when thermoviscous losses are incorporated is associated with the actuation frequency. The peak associated with the Helmholtz frequency is almost entirely dampened.

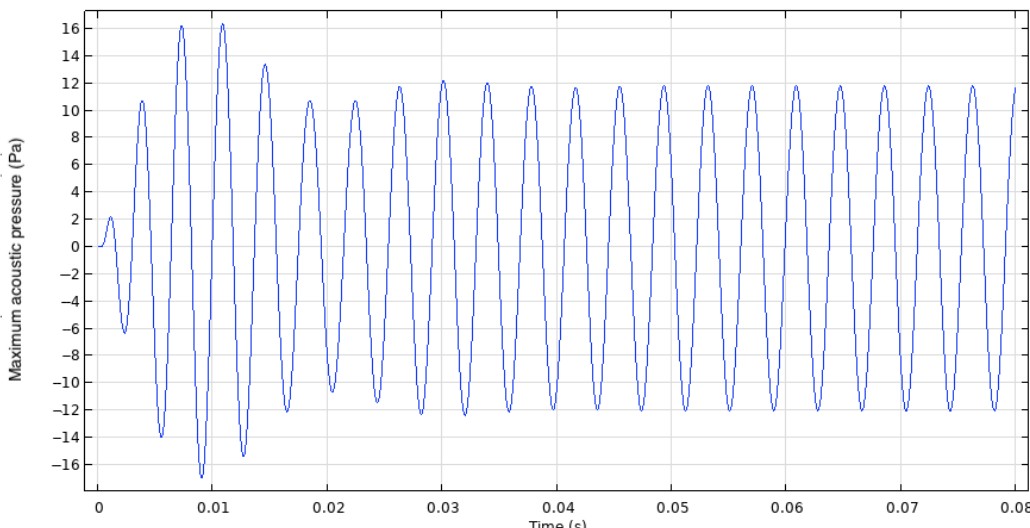

**Figure 25.** Transient acoustic response of synthetic jet actuator operating at 250 Hz with thermoviscous losses.

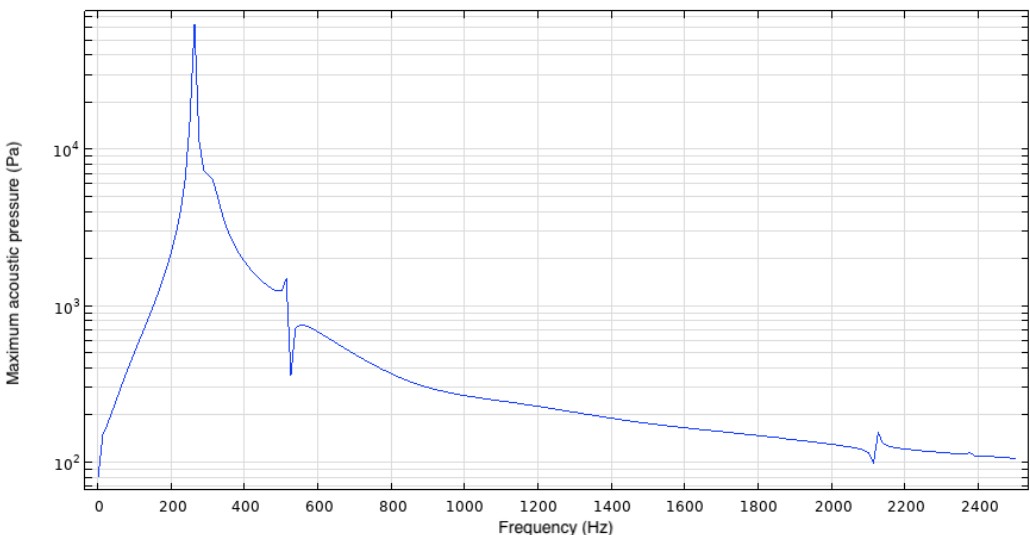

**Figure 26.** Discrete Fourier transform of the transient acoustic response of synthetic jet actuator operating at 250 Hz with thermoviscous losses.

The current work can also calculate the acoustic pressure in the far field. The perfectly matched layer and exterior boundary condition setup makes it possible to extrapolate to a point in space outside the computational domain to determine noise levels anywhere. The ability to study noise generation is an invaluable addition to the literature concerning SJAs. For designs to be viable in the real world, noise generation must be minimized. This work shows the importance of considering thermal and viscous losses when modeling and performing experiments. When performing experiments, extra care should be taken to standardize fluid properties and environmental conditions to prevent external viscous and thermal effects from impacting acoustic behavior. Furthermore, because the thermal and viscous boundary layers depend on frequency, these effects can help explain the acoustic response across a broad frequency range. For example, this explains the difference in dampening at the Helmholtz and mechanical resonant peaks.

### 3.3. Diaphragm

Many studies use an oscillating uniform velocity boundary condition instead of a moving wall for simplicity and computational efficiency [8,15,16,47]. Others wishing to

improve accuracy have modeled the diaphragm with an equation to describe the oscillating velocity profile [17,20]. Modeling the diaphragm as a moving wall with a profile defined by an equation has also been achieved with some success [19]. Some researchers have omitted the cavity and nozzle to simplify computations and focus on the exterior flow [48–50]. Some researchers have employed a more realistic multiphysics approach incorporating piezoelectricity and structural mechanics [27–30]. Figure 27 shows a comparison between the modeling methods that have been used, including the displacement profile determined through the current multiphysics modeling. These profiles assume the deflection is static, not operating in an active SJA. Apart from the clamped boundary condition, the second-order polynomial captures the deflection well [19] but does not account for the flexing the diaphragm undergoes while the SJA operates. The theoretical solution from [18] does not perform well for plates with multiple layers of different diameters.

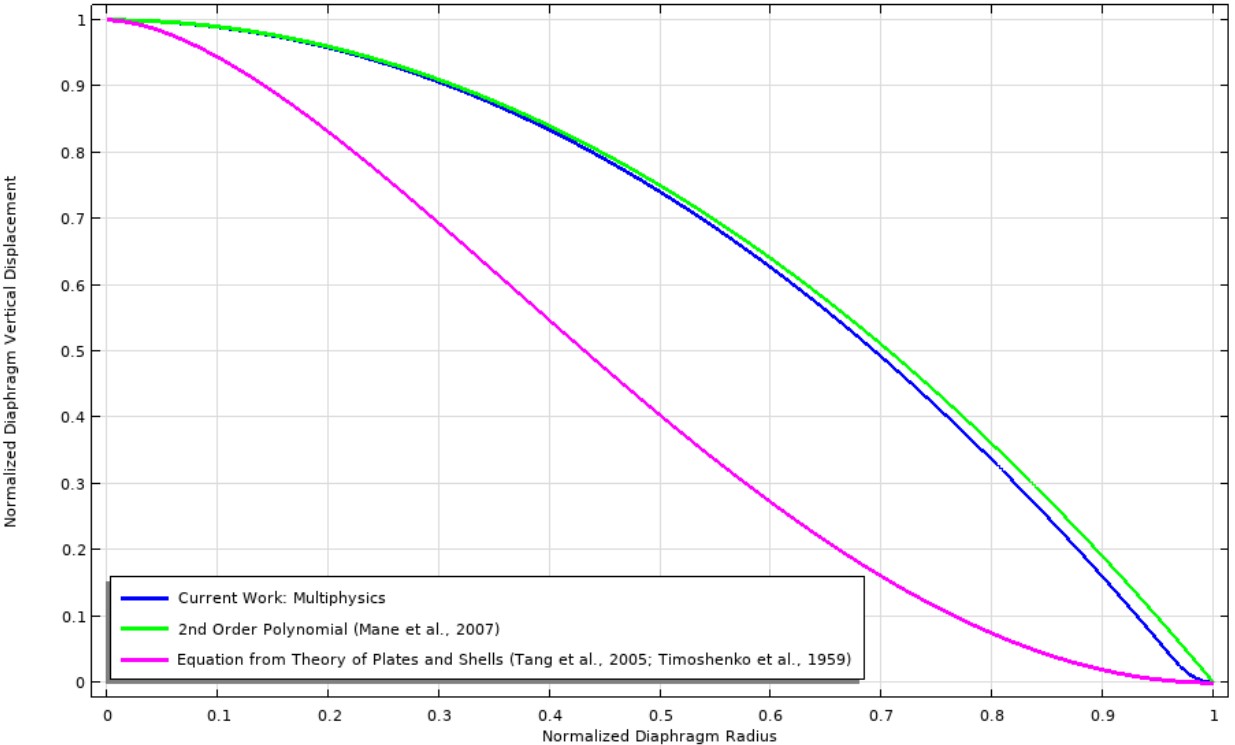

**Figure 27.** A comparison of the many diaphragm treatments in the literature to the current multiphysics approach [17–19].

When a static voltage is applied to a THUNDER TH-5C actuator, it deflects the same amount in each direction. However, when operating in an SJA in the presence of a flow field, the pressure in the SJA cavity varies sinusoidally and causes the diaphragm to flex. The force that this fluid pressure puts on the diaphragm as the vortex rings within the cavity impact its surface causes the diaphragm to deflect further downward (away from the cavity) rather than upward (toward the cavity). The impact that the fluid has on the diaphragm deflection is not insignificant. When operating near the Helmholtz frequency, the upper and lower maximum deflection difference is about 11.1%. Figure 28 shows the deflection of the SJA when operating at 260 Hz, normalized by maximum upward deflection. This indicates that approximating a diaphragm with an equation ignores the compliance of the diaphragm. This also shows that cavity compliance cannot be assumed to be zero, as some research suggests. The total volume of fluid displaced during an expulsion cycle when capturing the diaphragm shape using multiphysics and accounting for cavity compliance can be determined and compared against the approximations used by [17,19]. When normalized by peak diaphragm center displacement, the current work predicts the fluid volume displaced to be 6.1% and 38.7% more than found by [17,19], respectively.

For these reasons, modeling the diaphragm with multiphysics provides a more realistic representation of the boundary condition than a velocity profile or a moving wall.

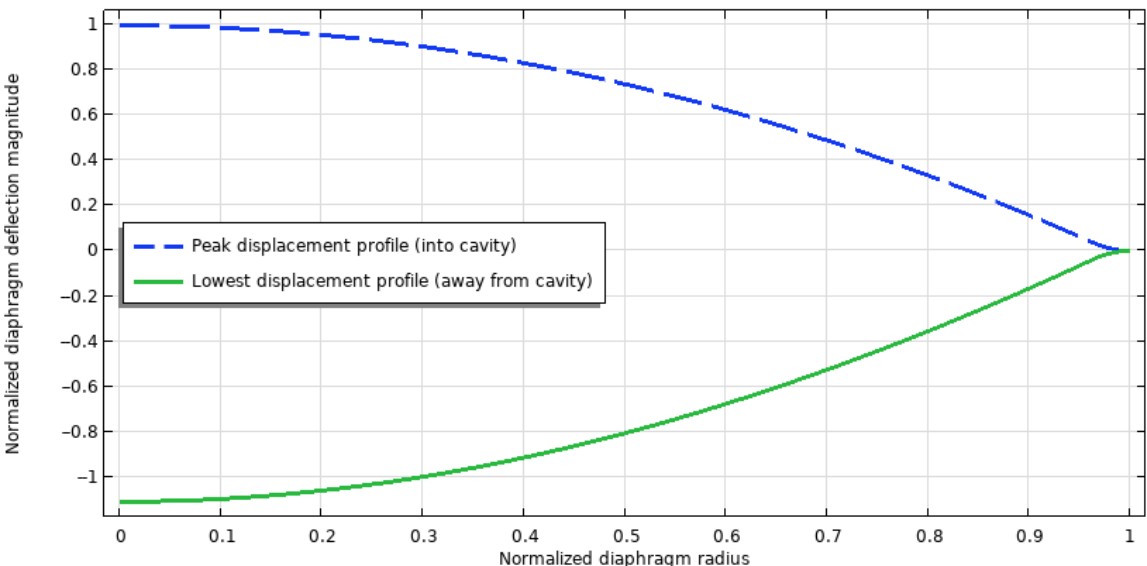

**Figure 28.** The deflection profiles of the diaphragm at its highest and lowest point while operating at 260 Hz.

## 4. Conclusions

A multiphysics approach was taken to model piezoelectric synthetic jet actuators. The model coupled physics from structural mechanics to provide insight into the combined flow, electrostatics, pressure acoustics, and fluid mechanics. The model was validated using hot-wire measurements and then used to study the fluid behavior, acoustic properties, and the impact of modeling a physical diaphragm. The model agreed with experimental work by capturing the Helmholtz frequency within 20 Hz and the maximum jet velocity within 0.8 m/s. Linear acoustics do not adequately model the acoustic effects with the current geometry, and thermoviscous effects are required to capture the viscous and thermal losses that occur in small geometries. It was shown that these losses dampen the fluid-borne noise associated with the cavity's resonance frequency, leaving the structure-borne noise related to the actuation frequency to be the dominant sound source.

**Author Contributions:** Conceptualization, M.G.M.B., A.E. and P.E.S.; methodology, M.G.M.B.; software, M.G.M.B.; validation, M.G.M.B.; formal analysis, M.G.M.B.; investigation, M.G.M.B.; resources, M.G.M.B., A.E. and P.E.S.; data curation, M.G.M.B.; writing—original draft preparation, M.G.M.B.; writing—review and editing, M.G.M.B., A.E. and P.E.S.; visualization, M.G.M.B.; supervision, A.E., P.E.S.; project administration, A.E. and P.E.S.; funding acquisition, A.E. and P.E.S. All authors have read and agreed to the published version of the manuscript.

**Funding:** This research was funded by Natural Sciences and Engineering Research Council of Canada (NSERC) grant number RGPIN-2022-03071, Dean's Strategic Fund, Dean's Fund: 2020-1, Canadian Microelectronics Corporation and the Digital Research Alliance of Canada (4752).

**Data Availability Statement:** The data presented in this study are available on request from the corresponding author.

**Conflicts of Interest:** The authors declare no conflicts of interest.

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
