# Peer review of "Multiphysics Modeling of a Synthetic Jet Actuator in Operation"

_actuators, doi:10.3390/act13020060_

Round 1
Reviewer 1 Report
Comments and Suggestions for Authors
See the attached file.

Author Response
Reviewer 1
We thank the reviewer for their helpful comments and believe they have improved the paper. Please find below a point-by-point reply to Reviewer 1.
- I didn’t see reference to Figure 1 in the paper. Could you check that?
Thank you for pointing this out, it has been corrected.
- Which numerical method is used in numerical simulation? Since COMSOL Multiphysics was used, this may imply that the finite element method was applied. Could you explain this in the paper? For example: which finite element spaces (Hdiv, H1 (continuous), L 2 (discontinuous)) are used to approximate each of the variables of the multiphysics problem?
Yes, the finite element method was used for the simulations. Most elements were “free” unstructured triangular elements, with a small portion being structured quadrilateral elements. The finite element spaces used depended on the variable being solved for in that domain. Here is a summary:
- Laminar flow: linear (1st-order) with Lagrange shape functions (P1).
- Solid mechanics: quadratic (2nd-order) elements with serendipity shape functions.
- Electrostatics: quadratic (2nd-order) elements.
- Pressure acoustics: quadratic (2nd-order) elements with Lagrange shape functions.
The default discretization/finite element space for both laminar and turbulent single-phase flow is linear elements with Lagrange shape functions.
2nd-order elements are important for the other physics as they are governed by PDEs with dominant 2nd derivatives.
A note about this has been made in the paper.
Reviewer 2 Report
Comments and Suggestions for Authors
This article makes a significant contribution to the acoustic modelling of the SJA.
Conclusions should be completed.
What is the influence of SJA geometry on the Helmholtz resonance frequency. The geometrical parameters of the SJA, such as the diameter of the orifice, the length of the orifice, and volume of the cavity, can change in a wide range.
An additional parameter that influences the Helmholtz resonance frequency of the SJA is the coupling ratio, which represents the ratio between the frequency of the pneumatic spring enclosed within the cavity and the natural frequency of the diaphragm.
Author Response
Reviewer 2
We thank the reviewer for their helpful comments and believe they have improved the paper. Please find below a point-by-point reply to Reviewer 2.
- Conclusions should be completed.
We have revised the paper and ensured all conclusions are provided.
- (a) What is the influence of SJA geometry on the Helmholtz resonance frequency. The geometrical parameters of the SJA, such as the diameter of the orifice, the length of the orifice, and volume of the cavity, can change in a wide range. (b)An additional parameter that influences the Helmholtz resonance frequency of the SJA is the coupling ratio, which represents the ratio between the frequency of the pneumatic spring enclosed within the cavity and the natural frequency of the diaphragm.
We appreciate your suggestion to explore more geometric parameters and coupling ratios. Our main objective in this study is to measure the effect of fluid and flow parameters on a fixed configuration that matches a well-documented experiment with a dynamic model that includes pressure feedback. We plan to extend our analysis to different configurations in future work.